# No $D_{\text{train}}$: Model-Agnostic Counterfactual Explanations Using Reinforcement Learning

**Xiangyu Sun**                                  *xiangyu.sun@rbcborealis.com*
*RBC Borealis*

**Raquel Aoki**                                  *raquel.aoki@rbcborealis.com*
*RBC Borealis*

**Kevin H. Wilson**                              *kevin.h.wilson@rbcborealis.com*
*RBC Borealis*

**Reviewed on OpenReview:** *https://openreview.net/forum?id=egNzAG9rOu*

## Abstract

Machine learning (ML) methods have experienced significant growth in the past decade, yet their practical application in high-impact real-world domains has been hindered by their opacity. When ML methods are responsible for making critical decisions, stakeholders often require insights into how to alter these decisions. Counterfactual explanations (CFEs) have emerged as a solution, offering interpretations of opaque ML models and providing a pathway to transition from one decision to another. However, most existing CFE methods require access to the model's training dataset, few methods can handle multivariate time-series, and none of model-agnostic CFE methods can handle multivariate time-series without training datasets. These limitations can be formidable in many scenarios. In this paper, we present NTD-CFE, a novel model-agnostic CFE method based on reinforcement learning (RL) that generates CFEs when training datasets are unavailable. NTD-CFE is suitable for both static and multivariate time-series datasets with continuous and discrete features. NTD-CFE reduces the CFE search space from a multivariate time-series domain to a lower dimensional space and addresses the problem using RL. Users have the flexibility to specify non-actionable, immutable, and preferred features, as well as causal constraints. We demonstrate the performance of NTD-CFE against four baselines on several datasets and find that, despite not having access to a training dataset, NTD-CFE finds CFEs that make significantly fewer and significantly smaller changes to the input time-series. These properties make CFEs more actionable, as the magnitude of change required to alter an outcome is vastly reduced. The code is available in the supplementary material.

## 1 Introduction

After receiving a negative decision—a denial of a loan, a poor performance review, or a rejection from a prestigious conference—a very natural question to ask is "What could I have done differently?" When the decision is made by a person, that question can be answered directly by the person. Indeed, peer reviews represent the reasons why a paper is accepted or rejected from a conference and, in the best case, give authors actionable feedback which may lead to an acceptance in the future (assuming the underlying decision algorithm remains unchanged). But when a decision is made or influenced by a black-box model, it can be much harder to provide insights. Telling a loan applicant they have a "poor credit score" does not tell them how they might approach getting approved at a later date.

Counterfactual Explanations (CFEs) (Wachter et al., 2017) were introduced to fill this gap. Given an input to a model, a CFE is perturbed version of the input which yields a prescribed output from the model. For

instance, if Bob's mortgage application is rejected, a CFE might suggest that Bob make $20,000 more per year, or alternatively, make $10,000 more per year and purchase a house in a different neighborhood.

CFEs also provide a method for examining how "fair" a model is, a concern that has become paramount in many real-world applications (Mehrabi et al., 2021; Angwin et al., 2022; Osoba et al., 2017). For instance, a CFE that shows if Bob's name were Alice his loan application would have been accepted points to potential discrimination on the basis of a protected characteristic. Classic models such as logistic regression and low-depth decision trees often called inherently interpretable (Verma et al., 2024; Murphy, 2012; Breiman, 2017) as the relative influence of input features can be read off from learned coefficients. But relations between learned parameters and input features are much harder to discern with more complex models. Indeed, even a logistic regression model with pairwise interaction terms can be complicated to reason about: Perhaps a woman who purchases a home in one neighborhood is less likely to have her application accepted than a man, but an additional $1,000 a year in income increases her chance of acceptance significantly more than a man's chance.

There exist many CFE methods for static datasets (Mothilal et al., 2020; Samoilescu et al., 2021; Verma et al., 2022). However, CFE methods for multivariate time-series data are less common due to the challenges posed by higher dimensionality (Ates et al., 2021; Theissler et al., 2022). Additionally, to the best of our knowledge, *all existing model-agnostic CFE methods for multivariate time-series require access to large collection of samples from the training distribution of the model being explained.* This requirement can be infeasible in real-world domains especially due to privacy and other concerns.

In this paper, we introduce No-Training-Dataset reinforcement-learning-based CFE (NTD-CFE), a reinforcement learning (RL) based CFE method designed for both static and multivariate time-series data containing both continuous and discrete attributes. Remarkably, NTD-CFE operates without training datasets or similar data samples and is model-agnostic so that it is compatible with any (even non-differentiable) predictive models. NTD-CFE also allows the user to specify which features they prefer to change, as well as how changing a particular feature may affect another feature, thus allowing the user to express both what counterfactuals are feasible for them and any causal relationships between those features. While NTD-CFE works for both static and time-series data, we focus on the harder setting of multivariate time-series data in this paper. We compare NTD-CFE to four state-of-the-art CFE methods on nine real-world multivariate time-series datasets. We find that NTD-CFE yields CFEs with significantly better proximity (the total magnitude of change proposed by the CFE) and sparsity (how many features the CFE proposes to alter) compared to the baselines.

The paper is structured as follows. We discuss related works and preliminaries in Section 2 and Section 3, respectively. Section 4 describes the proposed algorithm NTD-CFE. Experiments with nine real-world datasets and five predictive models are given in Section 5. Finally, we conclude in Section 6.

## 2 Related Works

**Explainable AI (XAI)** Counterfactual explanations belong to a much broader category of methods often called Explainable AI. XAI techniques can be broadly classified into two buckets (Verma et al., 2024): (a) inherently interpretable machine learning techniques and (b) post-hoc explanatory techniques. The first category is primarily a restriction on the types of models that a practitioner can employ to model a phenomenon. However, models often held out as interpretable (e.g., linear regression and low-depth decision trees (Murphy, 2012; Breiman, 2017)) may not have the capacity to capture complex phenomena. In the latter category, practitioners attempt to "model the model" with subsequent techniques (Sun et al., 2020; Liu et al., 2021). These methods can be further subdivided into global and local methods (Ates et al., 2021; Islam et al., 2021). Global methods attempt to simulate the opaque, complex logic of the original model using interpretable methods. On the other hand, local methods aim to explain the rationale behind a specific prediction made for a specific input. For instance, feature-based methods such as SHAP(Lundberg & Lee, 2017), identify features that have the most significant influence on a prediction. On the other hand, sample-based methods attempt to identify relevant samples to clarify a prediction (Kim et al., 2016; Mothilal et al., 2020). CFEs are an example of a post-hoc, local, sample-based explanation technique.

**Counterfactual Explanations (CFEs)** CFEs were introduced by Wachter et al. (2017) to explore an optimization-based technique for differentiable predictive models. Building upon this foundation, DiCE (Mothilal et al., 2020) noted that there were potentially many different CFEs proximal to any particular input and whether one was "better" than another was really a matter for the CFE's user to decide. As such, DiCE returns several diverse CFEs for any given sample. However, the base algorithm of DiCE is not guaranteed to return CFEs which satisfy causal constraints, and so DiCE introduced an expensive pruning step to filter "non-feasible" CFEs, i.e., those that do not satisfy causal constraints. These optimization-based methods are, unfortunately, not model-agnostic, as they require the underlying predictive model to be differentiable and thus exclude popular predictive models such as random forests and $k$-nearest neighbors. To overcome this limitation, methods such as those of Tsirtsis et al. (2021) leverage dynamic programming to identify an optimal counterfactual policy and subsequently utilize this policy to find CFEs. However, due to the high memory requirements of these methods, this technique is best suited for low-dimensional data with discrete features. Another line of work employs Bayesian optimization (BO) to address the CFE task (Spooner et al., 2021; Romashov et al., 2022; Huang et al., 2024). While these methods do not require training datasets, due to computational inefficiency of BO, they address static or univariate time-series data rather than multivariate time-series data.

**RL-based Methods** CFRL (Samoilescu et al., 2021) and FastAR (Verma et al., 2022) take a different tactic, employing techniques from RL to generate CFEs. CFRL first encodes samples into latent space using autoencoders (Kingma & Welling, 2013), then an RL agent is trained to find a CFE in the latent space. Finally, a decoder converts the latent CFE back to the input space. Similarly, FastAR converts the CFE problem to a Markov decision process (MDP) (Sutton & Barto, 2018) and then uses proximal policy optimization (PPO) (Schulman et al., 2017) to solve the MDP. However, the action candidates in FastAR are discrete and fixed, making it impractical for complex multivariate time-series data. Moreover, both CFRL and FastAR require access to training datasets.

**CFEs for Time-series Data** These and several other CFE techniques cater to static data, but methods for handling multivariate time-series data are much less prevalent (Theissler et al., 2022). For time-series data with $k$-nearest neighbor or random shapelet forest models, Karlsson et al. (2020) introduce one approach. Wang et al. (2021a) focus on univariate time-series, seeking CFEs in a latent space before decoding them back to the input space. Native-Guide (Delaney et al., 2021) finds the nearest unlike neighbor of an original univariate time-series sample, identifies the most influential subsequence of the neighbor, and substitutes it for the corresponding region in the original sample. On the other hand, CoMTE (Ates et al., 2021) can handle multivariate time-series data. First, it searches for distractor candidates, which are the original sample's neighbors in the training dataset that are predicted as the target class. Then, it identifies the best substitution parts on each distractor candidate. Finally, it gives a CFE by replacing a corresponding part of the original sample with the best substitution of the best distractor candidate.

**Adversarial Learning** We also note that, on the surface, generating adversarial examples seems quite similar to generating CFEs: both create proximal samples that yield distinct predictions from the original inputs. However, while adversarial learning considers proximity, essential CFE properties such as actionability, feasibility, and causal constraints are mostly ignored (Wachter et al., 2017; Verma et al., 2024; Sulem et al., 2022).

## 3 Preliminaries

**Problem Definition** CFEs aim to solve the following task: Given a user input $\boldsymbol{x}^*$, a predictive model $f$ and a target prediction $Y'$ such that $f(\boldsymbol{x}^*) \neq Y'$, the goal is to find a transformation from $\boldsymbol{x}^*$ to a new sample $\boldsymbol{x}'$ (i.e. CFE) such that (Verma et al., 2024; Karimi et al., 2020; Guidotti, 2022):

- valid: $f(\boldsymbol{x}') = Y'$,

- actionable: if the user cannot modify feature $j$, then $\boldsymbol{x}^*_j = \boldsymbol{x}'_j$,

- sparse: the CFE $\boldsymbol{x}'$ should differ in as few features from $\boldsymbol{x}^*$ as possible,

- proximal: the distance between $\boldsymbol{x}^*$ and $\boldsymbol{x}'$ should be small in some metric, and

- plausible: $\boldsymbol{x}'$ should satisfy all causal constraints on the input.

where $x_j$ denotes the $j$-th feature of $x$.

**Reinforcement Learning**   Our method for finding CFEs will rely on reinforcement learning (RL). In RL, an agent and an environment interact with each other (Sutton & Barto, 2018). The agent takes an action $a_t$ on a state $s_t$ at time step $t$. The environment receives $a_t$ and $s_t$ from the agent and returns the next state $s_{t+1}$ and a reward $R_{t+1}$ to the agent. The goal of the agent is to maximize the expected cumulative (discounted) reward. RL can be categorized as model-free RL and model-based RL. In model-free RL (Mnih et al., 2015; Haarnoja et al., 2018), the agent learns a policy from real experience when a model of the environment is not available to the agent. In model-based RL (Silver et al., 2017; Ha & Schmidhuber, 2018), the agent plans a policy from simulated experience generated by a model of the environment. This model of the environment is either learned or given. Furthermore, a policy-based RL algorithm (Williams, 1992; Schulman et al., 2017; 2015) typically samples actions from a policy network $\pi_{\boldsymbol{\theta}}$ parameterized by neural networks with parameters $\boldsymbol{\theta}$. Given a state, $\pi_{\boldsymbol{\theta}}$ is trained to return the best action that maximizes the expected cumulative reward.

## 4   The proposed method: NTD-CFE

In this work, we propose No-Training-Dataset reinforcement-learning-based CFE (NTD-CFE), formulating CFE as an RL problem. In this setup, the RL environment is the CFE predictive model $f$. The RL state $s$ is a sequence of CFEs beginning at the original user input $\boldsymbol{x}^*$ and ending in a final generated CFE $\boldsymbol{O^*}$. An action taken by the RL agent represents a small perturbation on the way from $\boldsymbol{x}^*$ to $\boldsymbol{O^*}$. A reward is a function of the predictive model and other objectives we introduce to maintain the properties discussed in Section 3 (more details below).

One-hot encoding is applied for categorical features. We assume that the prediction function of $f$ is computationally efficient, which is a common assumption in model-based RL (Sutton & Barto, 2018).

NTD-CFE pseudocode is given in Algorithm 1. Let $\boldsymbol{x}^* \in \mathbb{R}^{K \times D}$ denote a user input sample, where $K$ and $D$ denote the total number of time steps and features, respectively. To provide for plausibility, the $D$ features can optionally be categorized into actionable features $\boldsymbol{D}_{\text{act}}$, which the user can directly change; non-actionable features $\boldsymbol{D}_{\text{non-act}}$, which may change due to causal constraints but which the user cannot directly change (Sun & Schulte, 2023); and immutable features $\boldsymbol{D}_{\text{immu}}$, which may be used by the predictive model but which cannot change. $\boldsymbol{x}^*$ is static if $K = 1$ or temporal if $K > 1$. The time complexity of the algorithm is $O(M_E \cdot M_T)$.

**Action (Line 11 of Algorithm 1)**   NTD-CFE reduces the CFE search space from a multivariate time-series domain to a lower dimensional action space. Let $\pi_{\boldsymbol{\theta}}$ represent an RL policy network parameterized by neural networks with parameters $\boldsymbol{\theta}$. Each action $\boldsymbol{a}$ sampled from $\pi_{\boldsymbol{\theta}}$ is 3-dimensional $\boldsymbol{a} = \{a_{\text{time}}, a_{\text{feat}}, a_{\text{stre}}\}$, where $a_{\text{time}}$ denotes the time step of the intervention, $a_{\text{feat}}$ denotes which feature to intervene on, and $a_{\text{stre}}$ corresponds to the strength of the intervention. To be more specific, let $DC$ and $DD$ denote the numbers of actionable continuous and discrete features, respectively, where $DC + DD = |\boldsymbol{D}_{\text{act}}|$ and $|\boldsymbol{D}_{\text{act}}|$ denotes the total number of actionable features. Given an $\boldsymbol{x}$, the neural network parameterized by $\theta$ produces parameters to define four distributions $\{\boldsymbol{p}_{\text{time}}, \boldsymbol{p}_{\text{feat}}, \boldsymbol{\mu}_{\text{DC}}, \boldsymbol{\sigma}_{\text{DC}}, \boldsymbol{p}_{\boldsymbol{N}_{\text{dis}}}\} := \theta(\boldsymbol{x})$, such that $\boldsymbol{p}_{\text{time}} \in \mathbb{R}^K, \boldsymbol{p}_{\text{feat}} \in \mathbb{R}^{|\boldsymbol{D}_{\text{act}}|}, \boldsymbol{\mu}_{\text{DC}} \in \mathbb{R}^{DC}, \boldsymbol{\sigma}_{\text{DC}} \in \mathbb{R}^{DC}$ and $\boldsymbol{p}_{\boldsymbol{N}_{\text{dis}}} \in \mathbb{R}^{N_{\text{dis}}}$, where $N_{\text{dis}} = \sum_{i=1}^{DD} N_{\text{dis},i}$ and each $\boldsymbol{p}$ vector contains non-negative probabilities that sum to 1. The parameters $\boldsymbol{p}_{\text{time}}$ and $\boldsymbol{p}_{\text{feat}}$ define two categorical distributions from which $a_{\text{time}}$ and $a_{\text{feat}}$ are sampled, e.g. $a_{\text{time}} \sim \text{Cat}(\boldsymbol{p}_{\text{time}})$ and $a_{\text{feat}} \sim \text{Cat}(\boldsymbol{p}_{\text{feat}})$. When $a_{\text{feat}}$ is a continuous feature, the corresponding mean and standard deviation parameters $\mu_{\text{DC},a_{\text{feat}}}$ and $\sigma_{\text{DC},a_{\text{feat}}}$ define a normal distribution from which we sample how strong the intervention is for this continuous feature, e.g. $a_{\text{stre}} \sim N(\mu_{\text{DC},a_{\text{feat}}}, \sigma_{\text{DC},a_{\text{feat}}}^2)$. When $a_{\text{feat}}$ is a discrete feature, the corresponding parameters $\boldsymbol{p}_{\boldsymbol{N}_{\text{dis}},a_{\text{feat}}}$ define a categorical distribution from which we sample what the interventional value is for this discrete feature, e.g. $a_{\text{stre}} \sim \text{Cat}(\boldsymbol{p}_{\boldsymbol{N}_{\text{dis}},a_{\text{feat}}})$.

---

**Algorithm 1** NTD-CFE. Best viewed in color. Typical RL code is colored in gray.

---

1: **Input:** the original user input $\boldsymbol{x}^*$, a predictive model $f$, a target class $Y'$, a reward function $R$, a state transition function $F_p$, a proximity measure $D_{pxmt}$, a proximity weight $\lambda_{pxmt}$, feature feasibility weights $\boldsymbol{W}_{fsib}$, maximum number of episodes $M_E$, maximum number of interventions per episode $M_T$, discrete feature indicators $\boldsymbol{D}_{\mathrm{dis}}$, numbers of possible values of the discrete features $\{N_{\mathrm{dis},d}|d \in \boldsymbol{D}_{\mathrm{dis}}\}$.

2: **Optional Input:** non-actionable feature indicators $\boldsymbol{D}_{\mathrm{non\text{-}act}}$, immutable feature indicators $\boldsymbol{D}_{\mathrm{immu}}$, causal constraints $C_{causal}$, feature range constraints $C_{range}$, in-distribution detector $F_{\mathrm{in\_dist}}$, a discount factor $\gamma$, a learning rate $\alpha$, a regularization weight $\lambda_{WD}$

3: **Output:** a CFE $\boldsymbol{O}^*$

4: $\boldsymbol{O} = \{\emptyset\}$

5: $E := 0$

6: **while** $E < M_E$ **do**

7:     $\tau = \{\emptyset\}$  *# Keep a record of (state, action, reward) pairs*

8:     $t := 0$

9:     $\boldsymbol{x_t} := \boldsymbol{x}^*$

10:     **while** $t < M_T$ **do**

11:         $\boldsymbol{a_t} \sim \pi_{\boldsymbol{\theta}}(\cdot|\boldsymbol{x_t})$  *# Sample an action from the RL policy network*

12:         $\boldsymbol{x_{t+1}} := F_p(\boldsymbol{x_t}, \boldsymbol{a_t})$  *# State transition from the current $\boldsymbol{x}$ to the next $\boldsymbol{x}$*

13:         (Optionally, $\boldsymbol{x_{t+1}} := C_{range}(\boldsymbol{x_{t+1}}) \cdot C_{causal}(\boldsymbol{x_{t+1}}) \cdot \boldsymbol{D}_{\mathrm{immu}}(\boldsymbol{x_{t+1}}))$

14:         $r_{t+1} := R\left(f(\boldsymbol{x_{t+1}}), Y', D_{pxmt}(\boldsymbol{x}^*, \boldsymbol{x_{t+1}}, \boldsymbol{W}_{fsib}), \lambda_{pxmt}\right)$  *# Compute the reward*

15:         $\tau := \tau \cup (\boldsymbol{x_t}, \boldsymbol{a_t}, r_{t+1})$  *# Add the pair to the record*

16:         **if** $f(\boldsymbol{x_{t+1}}) = Y'$ and $\boldsymbol{x_{t+1}} \notin \boldsymbol{O}$ **then**  *# If a new valid CFE is found*

17:             $\boldsymbol{O} := \boldsymbol{O} \cup \boldsymbol{x_{t+1}}$ (Optionally, if also $F_{\mathrm{in\_dist}}(\boldsymbol{x_{t+1}}) = \mathrm{True}$)

18:             $t := t + 1$

19:             Break

20:         **end if**

21:         $t := t + 1$

22:     **end while**

23:     $T := t$

24:     **for** $t = 0, 1, \ldots, T - 1$ **do**  *# Update network parameters*

25:         $G := \sum_{t'=t+1}^{T} \gamma^{t'-t-1} \cdot r_{t'}$

26:         $\theta := \theta + \alpha \cdot \gamma^t \cdot G \cdot \nabla \ln \pi_{\boldsymbol{\theta}}(\boldsymbol{a_t}|\boldsymbol{x_t})$

27:     **end for**

28:     $E := E + 1$

29: **end while**

30: $\boldsymbol{O}^* := \min_i D_{pxmt}(\boldsymbol{x}^*, \boldsymbol{O_i}, \boldsymbol{W}_{fsib})$  *# Return the valid CFE with the lowest proximity*

---

**State Transition (Line 12 of Algorithm 1)** The state transition function $F_p$ can be any appropriate function for an application domain. In Section 5, we define $F_p(\boldsymbol{x_t}, \boldsymbol{a_t} = \{a_{\mathrm{time}}, a_{\mathrm{feat}}, a_{\mathrm{stre}}\})$ as:

$$
x_{t+1}^{\{k,d\}} := \begin{cases} x_t^{\{k,d\}} + a_{\mathrm{stre}} & \text{for } k \geq a_{\mathrm{time}} \text{ and } d = a_{\mathrm{feat}} \text{ (when feature } d \text{ is continuous)} \\ a_{\mathrm{stre}} & \text{for } k \geq a_{\mathrm{time}} \text{ and } d = a_{\mathrm{feat}} \text{ (when feature } d \text{ is discrete)} \\ x_t^{\{k,d\}} & \text{otherwise} \end{cases}
$$

where $x_t^{\{k,d\}}$ denotes the $d$-th feature of the $t$-th $\boldsymbol{x}$ at the time step $k$.

**Constraints (Line 13 of Algorithm 1)** Constraints (i.e., $C_{range}$ and $C_{causal}$) can be applied straightforwardly as sets of rules. For instance, a $C_{range}$ constraint can be a rule that enforces that Feature 1 must remain within range $[-1, 1]$. If an action attempts to change the value of Feature 1 to 2, the state transition can either be discarded or adjusted to cap the feature value at 1. Similarly, a $C_{causal}$ constraint such that Feature 1 is an interventional function of Feature 2, can be incorporated into the state transition function, ensuring that any action on Feature 2 updates Feature 1 accordingly. This dynamic is reflected in the reward function, which in turn influences the policy network, allowing the RL-based method to adapt accordingly.

**Reward and Proximity (Line 14 of Algorithm 1** ) We define the reward function $R$ as:

$$r := R\left(f(\boldsymbol{x}), Y', D_{pxmt}(\boldsymbol{x^*}, \boldsymbol{x}, \boldsymbol{W}_{fsib}), \lambda_{pxmt}\right) = \begin{cases} 1 - \lambda_{pxmt} \cdot D_{pxmt}(\boldsymbol{x^*}, \boldsymbol{x}, \boldsymbol{W}_{fsib}) & \text{if } f(\boldsymbol{x}) = Y' \\ 0 & \text{otherwise} \end{cases}$$

It combines a prediction reward (1 or 0) and a weighted proximity loss $D_{pxmt}$. $D_{pxmt}$ is 0 when $f(\boldsymbol{x}) \neq Y'$. Otherwise, in difficult settings where $f(\boldsymbol{x}) \neq Y'$ dominates over $f(\boldsymbol{x}) = Y'$, the RL agent would learn to produce CFEs that are too close to the original user input, which results in invalid CFEs. $\lambda_{pxmt}$ ensures that the reward is positive when $f(\boldsymbol{x}) = Y'$.

The proximity measure $D_{pxmt}$ can be any suitable measures for the application domain. In Section 5, we define $D_{pxmt}$ as the $L_1$-norm for continuous features and as the $L_0$-norm for discrete features, weighted by $\boldsymbol{W}_{fsib}$:

$$D_{pxmt}(\boldsymbol{x^d}, \boldsymbol{x'^d}, W_{fsib}^d) = \begin{cases} \sum_{k=1}^K |x^{\{k,d\}} - x'^{\{k,d\}}| \cdot W_{fsib}^d & \text{(if feature } d \text{ is continuous)} \\ \sum_{k=1}^K I(x^{\{k,d\}} \neq x'^{\{k,d\}}) \cdot W_{fsib}^d & \text{(if feature } d \text{ is discrete)} \end{cases} \tag{1}$$

$W_{fsib}^d$ denotes the feasibility to change the $d$-th feature, which encodes the user's preference on altering this feature. $\boldsymbol{x}^d$ denotes the $d$-th feature of $\boldsymbol{x}$, and $x^{\{k,d\}}$ denotes the $d$-th feature of $\boldsymbol{x}$ at the time step $k$. NTD-CFE prefers to generate CFEs by altering features associated with small $W_{fsib}$. If a user does not specify preference on features, $\boldsymbol{W}_{fsib}$ is 1 for all features.

**Classification and Regression (Line 16 of Algorithm 1)** Algorithm 1 describes NTD-CFE for a classification model $f$. As a model-agnostic method, NTD-CFE supports not only classification but also regression models. To work with a regression predictive model $f$, one can replace the first condition of Line 16 by $Y'_{lower} \leq f(\boldsymbol{x_{t+1}}) \leq Y'_{upper}$, where $Y'_{lower}$ and $Y'_{upper}$ represent the lower and upper bounds for the target regression value, respectively.

**Output (Lines 17 and 30 of Algorithm 1)** On Line 17, if a valid CFE is reached and is not already in the set $\boldsymbol{O}$, then it is added to $\boldsymbol{O}$. Optionally, an additional condition can be imposed such that a valid CFE is added to $\boldsymbol{O}$ only if it is also plausible. Finally, on Line 30, NTD-CFE returns the CFE with the lowest proximity from the set $\boldsymbol{O}$.

**Operating Without Training Datasets** The input $\boldsymbol{x}^*$ in Algorithm 1 is a testing sample $X_{\text{test}}$, i.e., the user sample to be explained or for which CFEs are generated. To clarify, the baseline methods require a training dataset $D_{\text{train}}$ to generate CFEs for $X_{\text{test}}$, whereas the proposed method operates directly on $X_{\text{test}}$ without the need for $D_{\text{train}}$. More details about how the baselines use $D_{\text{train}}$ is discussed in Section 2.

**Motivation of Using RL** A search algorithm is required to solve the model-agnostic CFE problem without a training dataset. For multivariate time-series data, the CFE search space $(K \times D)$ is large. BO-based CFE methods primarily focus on static or univariate time-series data (Spooner et al., 2021; Romashov et al., 2022; Huang et al., 2024) and do not scale well to multivariate time series. Optimization-based approaches, such as DiCE, are not model-agnostic since they require differentiable predictive models. Instead, we formulate the search problem as a continuous control task using reinforcement learning (RL), reducing it to a lower dimensional action space for improved scalability. Additionally, causal constraints are crucial for CFE methods (Verma et al., 2024). While BO-based methods can impose simple value constraints (e.g., bounding ranges), they struggle with complex causal constraints. Our RL approach encodes causal constraints more naturally.

## 4.1 Limitation

Features on different scales can impact the performance of models like neural networks. Without training datasets, standardizing the features becomes impractical. One approach is to leverage $W_{fsib}$ to mitigate the impact of continuous features on different scales. Assuming domain knowledge about the ranges of feature values (which is often available in practice), one can assign smaller $W_{fsib}$ to continuous features on

larger scales and larger $W_{fsib}$ to continuous features on smaller scales. In our evaluation, we assume that the continuous features have a mean of 0 and a variance of 1. We leave the evaluation of this approach or a more sophisticated approach for future work.

The performance of the proposed model is unaffected by the size of the predictive model. Instead, its performance depends on the frequency with which the predictive model returns the desired prediction. We assume that the desired prediction returned by the predictive model is not sparse. This is a realistic assumption in many real-world domains. For example, numerous applicants with diverse personal characteristics applying for credit cards receive approval from the decision model of a bank. This assumption ensures that the RL agent receives enough reward signals to improve its performance. Without both training datasets and reward signals, it would be extremely challenging for ML methods to solve meaningful tasks.

## 5 Evaluation

In this section, we provide qualitative examples and quantitative experiment results to demonstrate the effectiveness of NTD-CFE for multivariate data-series data. The details about datasets and hyperparameters are provided in Appendix A and Appendix C, respectively.

### 5.1 Qualitative Examples

We illustrate qualitative examples generated by NTD-CFE with two interpretable rule-based predictive models and an interpretable Life Expectancy dataset. Please refer to Appendix B.1 for the definitions of the interpretable rule-based models. All examples in this section are generated using the first sample, which represents the country Albania, in the Life Expectancy dataset.

#### 5.1.1 Equal feature feasibility weights $W_{fsib}$

In Figure 1, all features have equal feasibility weights ($W_{fsib} = 1.0$) as no user preferences are set. In Figure 1a, following the definition of rule-based model 1 (Appendix B.1), Albania's prediction is 0 (i.e. undesired), because "*GDP-per-capita*" *and* "*health-expenditure*" in the last 5 years are below 0. Accordingly, NTD-CFE generates a CFE by increasing these values above 0. In Figure 1b, if at least one of "*GDP-per-capita*" *or* "*health-expenditure*" is above 0 in the last 5 years, then rule-based model 2 predicts 1 (i.e. desired). NTD-CFE generates a valid CFE for rule-based model 2 by raising "*GDP-per-capita*" above 0.

#### 5.1.2 Different feature feasibility weights $W_{fsib}$

However, changing "*GDP-per-capita*" for Albania may be impractical. An alternative way to make rule-based model 2 predict 1 is to increase "*health-expenditure*" above 0. NTD-CFE can achieve this in three different ways: (1) marking "*GDP-per-capita*" as non-actionable; (2) assigning a small feasibility weight to "*health-expenditure*;*" or (3) assigning a large feasibility weight to "*GDP-per-capita*." (1) is straightforward with NTD-CFE. Hence, we only present the results for (2) and (3). In Appendix Figure 2a, the feasibility weight $W_{fsib}$ for "*health-expenditure*" is set to 0.1, ten times smaller than that of "*GDP-per-capita*," which remains unchanged as 1. With the reduced feasibility weight for "*health-expenditure*," NTD-CFE generates a valid CFE by modifying "*health-expenditure*." In Appendix Figure 2b, the feasibility weight for "*GDP-per-capita*" is set to 10, ten times greater than other features. With the high feasibility weight for "*GDP-per-capita*," NTD-CFE preserves "*GDP-per-capita*" and looks for other features to achieve the desired prediction. As a result, NTD-CFE learns to alter "*health-expenditure*".

Next, we show that setting small feasibility weights on irrelevant features does not affect the CFEs generated by NTD-CFE. In Appendix Figure 2c, although the feasibility weights for irrelevant features "*CO2-emissions*," "*electric-power-consumption*," and "*forest-area*" are set to be ten times smaller than others, NTD-CFE still alters the relevant feature "*GDP-per-capita*". Additional results for different feasibility weights are provided in Appendix Figure 3.

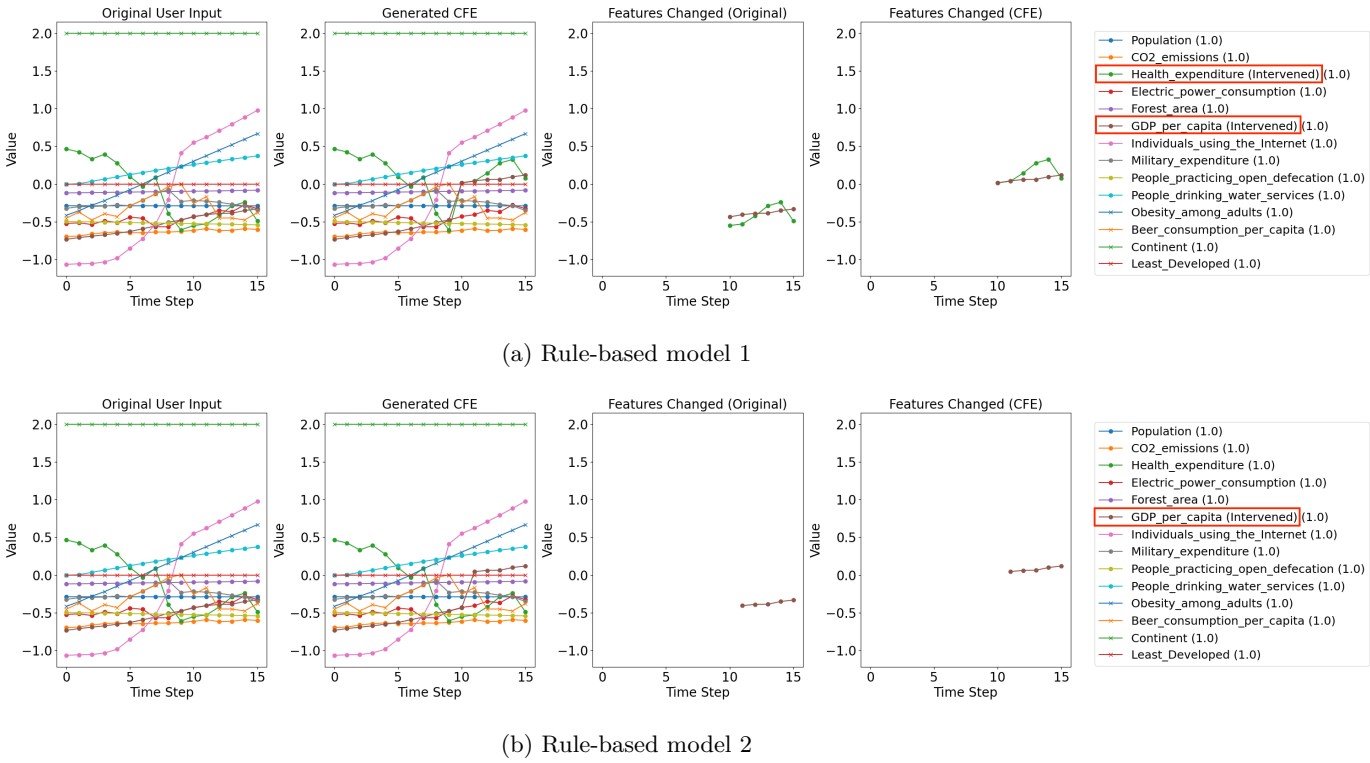

(a) Rule-based model 1

(b) Rule-based model 2

Figure 1: Qualitative examples with rule-based models and equal feature feasibility weights $W_{fsib}$. "Original User Input" shows the original input with the original feature values. "Generated CFE" shows the generated CFE with the modified features. The other two plots, "Features Changed (Original)" and "Features Changed (CFE)", omit most features and only show the modified features.

## 5.2 Quantitative Experiments

In this section, we compare NTD-CFE to four baseline methods in 45 experiments, which correspond to nine real-world datasets each evaluated with five predictive models. Additional experiments are provided and analyzed in the appendix.

**Datasets** Nine real-world multivariate time-series datasets are used for evaluation (Appendix A for details).

**Baseline Methods** We benchmark NTD-CFE against four model-agnostic baseline methods: CoMTE (Ates et al., 2021), Native-Guide (Delaney et al., 2021), CFRL (Samoilescu et al., 2021), and FastAR (Verma et al., 2022). Optimization-based methods (Mothilal et al., 2020; Sulem et al., 2022; Hsieh et al., 2021) are excluded from the comparison, because our predictive models are not restricted to differentiable models. For Native-Guide, we follow the approach of Bahri et al. (2022) by concatenating multivariate time-series samples into univariate time-series samples. Unlike the proposed NTD-CFE, the baselines require training datasets, i.e. either $(\boldsymbol{X}_{train}, \boldsymbol{Y}_{train})$ or $(\boldsymbol{X}_{train})$. We omit comparisons with other methods and use these popular methods as the representative baselines.

**Predictive Models** Five predictive models are employed per dataset: a long short-term memory (LSTM) neural network, a k-nearest neighbor (KNN), a random forest and two interpretable rule models; see Appendix B for details. Given nine datasets, there is a total of 45 predictive models.

Table 1: Quantitative results with the eRing dataset for different predictive models and methods. Our proposed method NTD-CFE consistently achieves significantly better proximity and sparsity.

| Predictive Model | $N_{inv}$ | Methods | Success Rate (%) | Validity Rate (%) | Plausibility Rate (%) | Proximity | Sparsity |
|---|---|---|---|---|---|---|---|
| LSTM | 14 | CoMTE | **100** | **100** | **100** | 346.746 | 260.0 |
| | | Native-Guide | **100** | **100** | **100** | 316.502 | 260.0 |
| | | CFRL | 0 | — | — | — | — |
| | | FastAR | 0 | — | — | — | — |
| | | **NTD-CFE** | **100** | **100** | **100** | **54.682** | **31.214** |
| KNN | 16 | CoMTE | **100** | **100** | **100** | 338.141 | 260.0 |
| | | Native-Guide | **100** | **100** | 93.75 | $3.789 \times 10^{11}$ | 229.938 |
| | | CFRL | **100** | **100** | **100** | 321.046 | 260.0 |
| | | FastAR | 0 | — | — | — | — |
| | | **NTD-CFE** | **100** | **100** | **100** | **144.937** | **86.688** |
| Random Forest | 15 | CoMTE | **100** | **100** | **100** | 340.338 | 260.0 |
| | | Native-Guide | **100** | **100** | 93.333 | $3.275 \times 10^{12}$ | 246.467 |
| | | CFRL | 0 | — | — | — | — |
| | | FastAR | 0 | — | — | — | — |
| | | **NTD-CFE** | **100** | **100** | **100** | **68.882** | **46.733** |
| Rule Based 1 | 29 | CoMTE | 0 | — | — | — | — |
| | | Native-Guide | **100** | **100** | 44.828 | $9.928 \times 10^{14}$ | 256.552 |
| | | CFRL | 0 | — | — | — | — |
| | | FastAR | 0 | — | — | — | — |
| | | **NTD-CFE** | **100** | **100** | **100** | **103.953** | **63.345** |
| Rule Based 2 | 25 | CoMTE | **100** | **100** | **100** | 347.295 | 260.0 |
| | | Native-Guide | **100** | **100** | 88.0 | $5.075 \times 10^{13}$ | 245.4 |
| | | CFRL | 0 | — | — | — | — |
| | | FastAR | 0 | — | — | — | — |
| | | **NTD-CFE** | **100** | **100** | **100** | **31.301** | **14.76** |

### 5.2.1  Results

The methods are evaluated using five metrics. Let $N_{inv}$ denote the total number of invalid samples, i.e., those classified as the undesired class by a predictive model, $N_{inv\_val}$ denote the number of invalid samples for which a CFE method generates valid CFEs, $N_{val}$ denote the number of valid CFEs generated by a CFE method, $N_{CFE}$ denote the number of CFEs generated by a CFE method, and $N_{plau\_val}$ denote the number of plausible and valid CFEs generated by a CFE method. We set feature feasibility weights $W_{fsib}^d = 1$ for all the features $d \in D$.

Table 1 shows the results with the eRing dataset as an example. The proposed NTD-CFE consistently has the lowest proximity and sparsity. The results for all other datasets are given in Appendix Tables 3 to 10. In this section, we analyze and compare the complete results, including these in the appendix.

Results dimmed in gray in the tables are skipped from analysis and comparison. For Plausibility Rate, Proximity and Sparsity, we only compare the methods under 100% success rates. The reason is that if a method always generates CFEs that are close to the original invalid user input $\boldsymbol{x}^*$, even though these CFEs may often be invalid due to their closeness to $\boldsymbol{x}^*$, the plausibility rate, proximity and sparsity will always be superior. In the extreme case, if a method always returns the original but invalid $\boldsymbol{x}^*$, it would achieve a perfect plausibility rate, proximity and sparsity, but at the same time fail completely as a CFE method in terms of success rate.

**Success Rate:** $\frac{N_{inv\_val}}{N_{inv}}$ There are two scenarios for a CFE method to fail: 1) no valid CFEs are generated; 2) no CFEs (either valid or invalid) are generated. For RL-based baselines, CFRL and FastAR fail with a 0%

success rate in 30/45 and 34/40 cases (excluding the 5 cases where FastAR crashes due to memory usage on Table 4), respectively. NTD-CFE outperforms CFRL in 31/45 cases, and is on par with CFRL in 10/45 cases. In contrast, NTD-CFE underperforms CFRL in 4/45 cases. NTD-CFE outperforms FastAR in all 40/40 cases. Compared to the other baselines, Native-Guide, CoMTE and NTD-CFE fail with a 0% success rate in 0/40 (excluding the 5 cases where Native-Guide crashes due to memory usage on Table 4), 3/45 and 1/45 cases, respectively. NTD-CFE outperforms Native-Guide in 8/40 cases, and is on par with it in 25/40 cases. NTD-CFE underperforms Native-Guide in 7/40 cases. However, the minimum success rate of Native-Guide is 30.855%, which is better than that of NTD-CFE (0.68%). NTD-CFE underperforms CoMTE in success rate. NTD-CFE produces lower success rates than CoMTE in 15/45 cases, achieves the same success rates in 27/45 cases, and outperforms it in only 3/45 cases. More comparison with CoMTE is provided at the end of this subsection.

It is important to note that: 1) We provide training datasets to the baselines (because they require training datasets to operate), but not to NTD-CFE. This additional information provided only to the baselines gives them an advantage over NTD-CFE. Without training datasets, the methods stop working except NTD-CFE. 2) In Appendix Table 19 we show that the success rate of NTD-CFE can be further improved, e.g. from 0.68% to 76.87%.

**Validity Rate:** $\frac{N_{val}}{N_{CFE}}$   Both NTD-CFE and CoMTE ensure perfect validity rates by design (i.e. 100%); they either produce a valid CFE or do not produce a CFE at all. In contrast, the other three baselines may return invalid CFEs; therefore, their validity rates can be lower than 100%. Therefore, NTD-CFE and CoMTE outperform other baselines in terms of validity rate.

**Plausibility Rate:** $\frac{N_{plau\_val}}{N_{val}}$   The comparisons to CFRL and FastAR are skipped because more than half of the experiments yield 0% success rates, and therefore, undefined plausibility rates. NTD-CFE outperforms and is on par with Native-Guide in 14/24 and 1/24 cases, respectively. NTD-CFE underperforms Native-Guide in 9/24 cases. NTD-CFE is on par with CoMTE in 8/27 cases and underperforms it in 19/27 cases. In summary, in terms of plausibility rate, CoMTE outperforms the proposed NTD-CFE, and NTD-CFE outperforms Native-Guide. Again, the baselines have the advantage by utilizing additional training information that is not provided to NTD-CFE.

Additionally, one can enforce plausibility in NTD-CFE (Line 17 of Algorithm 1). NTD-CFE achieves 100% plausibility rates at the cost of lower success rates and higher proximity and sparsity. Please see Appendix D for details.

**Proximity and Sparsity**   Proximity and Sparsity are defined as the $L_1$-norm or $L_0$-norm, respectively, of the difference between a CFE and the original $\boldsymbol{x}^*$ (Verma et al., 2022; Samoilescu et al., 2021). Due to the aforementioned reason, proximity and sparsity are computed only with valid CFEs. Therefore, comparison with FastAR is skipped. *NTD-CFE outperforms all baselines in proximity and sparsity in all cases. It also surpasses the baselines by a large margin.* For example, there are 4, 8 or 15 cases where the proximity of NTD-CFE is at least 20 times, 10 times or 5 times lower than that of all the baselines, respectively (e.g., 16.183 vs. 220.552 in Appendix Table 7). Similarly, there are 3, 5 or 21 cases where the sparsity of NTD-CFE is at least 50 times, 20 times or 10 times lower than that of all the baselines, respectively (e.g., 20.587 vs. 1224.0 in Appendix Table 5).

**Comparison with RL-based methods.**   *NTD-CFE outperforms the two RL-based baselines, CFRL and FastAR, in all the metrics.* CFRL and FastAR often fail to generate valid CFEs for complex multivariate time-series data. Please note that this is not a criticism of CFRL or FastAR, because they are not designed for multivariate time-series data.

**Comparison with CoMTE**   Although CoMTE outperforms NTD-CFE in success rate and in plausibility rate, it is important to highlight that: 1) *CoMTE requires a training dataset, while NTD-CFE does not.* The better performance of CoMTE over NTD-CFE comes at the cost of needing more information and reduced versatility in practical applications. 2) CoMTE relies on finding distractors correctly classified as the target class. Tables 1, 5 and 9 for rule-based model 1 show that when the predictive models classify all training

samples as the undesired class, CoMTE fails completely with a 0% success rate. In contrast, NTD-CFE is more versatile and can operate in such difficult situations. 3) Appendix Table 19 shows that the success rates of NTD-CFE can be improved by increasing the maximum number of episodes $M_E$ or the maximum number of interventions per episode $M_T$, which correspond to more exhaustive RL search.

## 6 Conclusion

In this paper, we introduce NTD-CFE, a model-agnostic RL-based method that generates CFEs for static and multivariate time-series data. NTD-CFE operates without requiring a training dataset, is compatible with both classification and regression predictive models, handles continuous and discrete features, and offers functionalities such as feature feasibility (i.e. user preference), feature actionability and causal constraints. We illustrate the effectiveness of NTD-CFE through qualitative examples and benchmark it against four state-of-the-art methods using nine real-world multivariate time-series datasets. Our results consistently show that NTD-CFE produces CFEs with significantly better proximity and sparsity. Future research includes extending the work to large language models, examining more advanced RL algorithms to potentially improve performance, using $W_{fsib}$ for features on different scales, and exploring alternation solutions other than RL for multivariate time-series data without training datasets.

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

## A Datasets

Nine real-world multivariate time-series datasets are used for evaluation in Section 5.2:

**Life Expectancy** [1]   Each sample has 16 time steps (from 2000 to 2015) and 17 features per time step. All the features are interpretable. Please see Table 2 for feature names and types. We remove "*Country Name*" and "*Year*" from the list of input features and use "*Life Expectancy*" in 2015 as the label. Therefore, the dataset has $K = 16$ and $D = 14$ in our notation. We set $Y = 1$ if "*Life Expectancy*" in 2015 is greater or equal to 75 as the target class and otherwise $Y = 0$ as the undesired class.

**NATOPS** [2]   The NATOPS dataset contains sensory data on hands, elbows, wrists and thumbs to classify movement types. $K = 51$ and $D = 24$. All the features in this dataset are continuous. There are 6 classes of different movements. We set class 4 to 6 as the target class.

**PEMS-SF** [3]   $K = 144$ and $D = 963$. The dataset contains transportation data from California. All the features are continuous. There are seven classes and we use the last four classes as the target class.

**Heartbeat** [4]   $K = 405$ and $D = 61$. All the features in this dataset are continuous. There are two classes: *normal* heartbeat (target class) and *abnormal* heartbeat (undesired class).

**eRing**[5]   $K = 65$ and $D = 4$. All the features in this dataset are continuous. There are six classes and we use the last three classes as the target class.

**Racket Sports** [6]   $K = 30$ and $D = 6$. All the features in this dataset are continuous. There are four classes: "*Badminton Smash*", "*Badminton Clear*", "*Squash Forehand Boast*" and "*Squash Backhand Boast*". We use the last two classes as the target class.

**Basic Motions** [7]   $K = 100$ and $D = 6$. All the features in this dataset are continuous. There are four classes: "*Badminton*", "*Running*", "*Standing*" and "*Walking*". We use "*Standing*" as the target class.

**Japanese Vowels** [8]   $K = 29$ and $D = 12$. All the features in this dataset are continuous. There are nine classes and we use the last five classes as the target class.

**Libras** [9]   $K = 45$ and $D = 2$. All the features in this dataset are continuous. There are 15 classes and we use the last eight classes as the target class.

All categorical features are one-hot encoded. All continuous features are standardized to have mean 0 and variance 1.

## B  Predictive Models Used in Section 5.2

Since NTD-CFE is model-agnostic, we expect it to work for predictive models with arbitrary hyperparameter values and architectures. In Section 5.2, we evaluate the methods using five predictive models: two interpretable rule- based models, a long short-term memory (LSTM) neural network, a K-nearest neighbor (KNNs), and a random forest.

---

[1]https://www.kaggle.com/datasets/vrec99/life-expectancy-2000-2015
[2]http://www.timeseriesclassification.com/description.php?Dataset=NATOPS
[3]https://www.timeseriesclassification.com/description.php?Dataset=PEMS-SF
[4]http://www.timeseriesclassification.com/description.php?Dataset=Heartbeat
[5]https://www.timeseriesclassification.com/description.php?Dataset=ERing
[6]https://www.timeseriesclassification.com/description.php?Dataset=RacketSports
[7]https://www.timeseriesclassification.com/description.php?Dataset=BasicMotions
[8]https://www.timeseriesclassification.com/description.php?Dataset=JapaneseVowels
[9]https://www.timeseriesclassification.com/description.php?Dataset=Libras

### B.1 Rule-based predictive models

Two interpretable rule-based predictive models are implemented for each dataset.

**Rule-based models for the Life Expectancy dataset**

We employed two interpretable rule-based predictive models for the Life Expectancy dataset in Section 5.1 and Section 5.2. Let $d_1$, $d_2$, $d_3$, $d_4$, $d_5$ denote the features "*least-developed*", "*GDP-per-capita*", "*health-expenditure*", "*people-using-at-least-basic-drinking-water-services*" and "*people-practicing-open-defecation*", respectively. We define the first rule-based model as:

$$f(\boldsymbol{x}) = \begin{cases} Y' & \text{if } x^{\{k,d_1\}} = 0 \wedge x^{\{k,d_2\}} > 0 \wedge x^{\{k,d_3\}} > 0 \wedge x^{\{k,d_4\}} > 0 \wedge x^{\{k,d_5\}} < 0 \text{ for } K - 4 \leq k \leq K \\ Y & \text{otherwise} \end{cases}$$

and the second rule-based model as:

$$f(\boldsymbol{x}) = \begin{cases} Y' & \text{if } x^{\{k,d_1\}} = 0 \wedge \left( x^{\{k,d_2\}} > 0 \vee x^{\{k,d_3\}} > 0 \right) \wedge x^{\{k,d_4\}} > 0 \wedge x^{\{k,d_5\}} < 0 \text{ for } K - 4 \leq k \leq K \\ Y & \text{otherwise} \end{cases}$$

where $Y'$ is the target class and $Y$ is the undesired class.

**Rule-based models for the PEMS-SF dataset**

Let $d_i$ denote the $i$-th feature. We define the first rule-based model as:

$$f(\boldsymbol{x}) = \begin{cases} Y' & \text{if } x^{\{k,d_1\}} > 0 \wedge x^{\{k,d_{100}\}} > 0 \wedge x^{\{k,d_{300}\}} > 0 \text{ for } K - 50 \leq k \leq K \\ Y & \text{otherwise} \end{cases}$$

and the second rule-based model as:

$$f(\boldsymbol{x}) = \begin{cases} Y' & \text{if } x^{\{k,d_1\}} > 0 \vee x^{\{k,d_{100}\}} > 0 \vee x^{\{k,d_{300}\}} > 0 \text{ for } K - 50 \leq k \leq K \\ Y & \text{otherwise} \end{cases}$$

where $Y'$ is the target class and $Y$ is the undesired class.

**Rule-based models for the NATOPS dataset**

Let $d_1$ and $d_2$ denote the features "*Hand tip left, X coordinate*" and "*Hand tip right, X coordinate*", respectively. We define the first rule-based model as:

$$f(\boldsymbol{x}) = \begin{cases} Y' & \text{if } x^{\{k,d_1\}} > 0 \wedge x^{\{k,d_2\}} > 0 \text{ for } K - 9 \leq k \leq K \\ Y & \text{otherwise} \end{cases}$$

and the second rule-based model as:

$$f(\boldsymbol{x}) = \begin{cases} Y' & \text{if } x^{\{k,d_1\}} > 0 \vee x^{\{k,d_2\}} > 0 \text{ for } K - 9 \leq k \leq K \\ Y & \text{otherwise} \end{cases}$$

where $Y'$ is one of the target classes and $Y$ is one of the undesired classes.

**Rule-based models for the Heartbeat dataset**

Let $d_i$ denote the $i$-th feature. We define the first rule-based model as:

$$f(\boldsymbol{x}) = \begin{cases} Y' & \text{if } x^{\{k,d_1\}} > 0 \wedge x^{\{k,d_2\}} > 0 \wedge x^{\{k,d_3\}} > 0 \text{ for } K - 4 \leq k \leq K \\ Y & \text{otherwise} \end{cases}$$

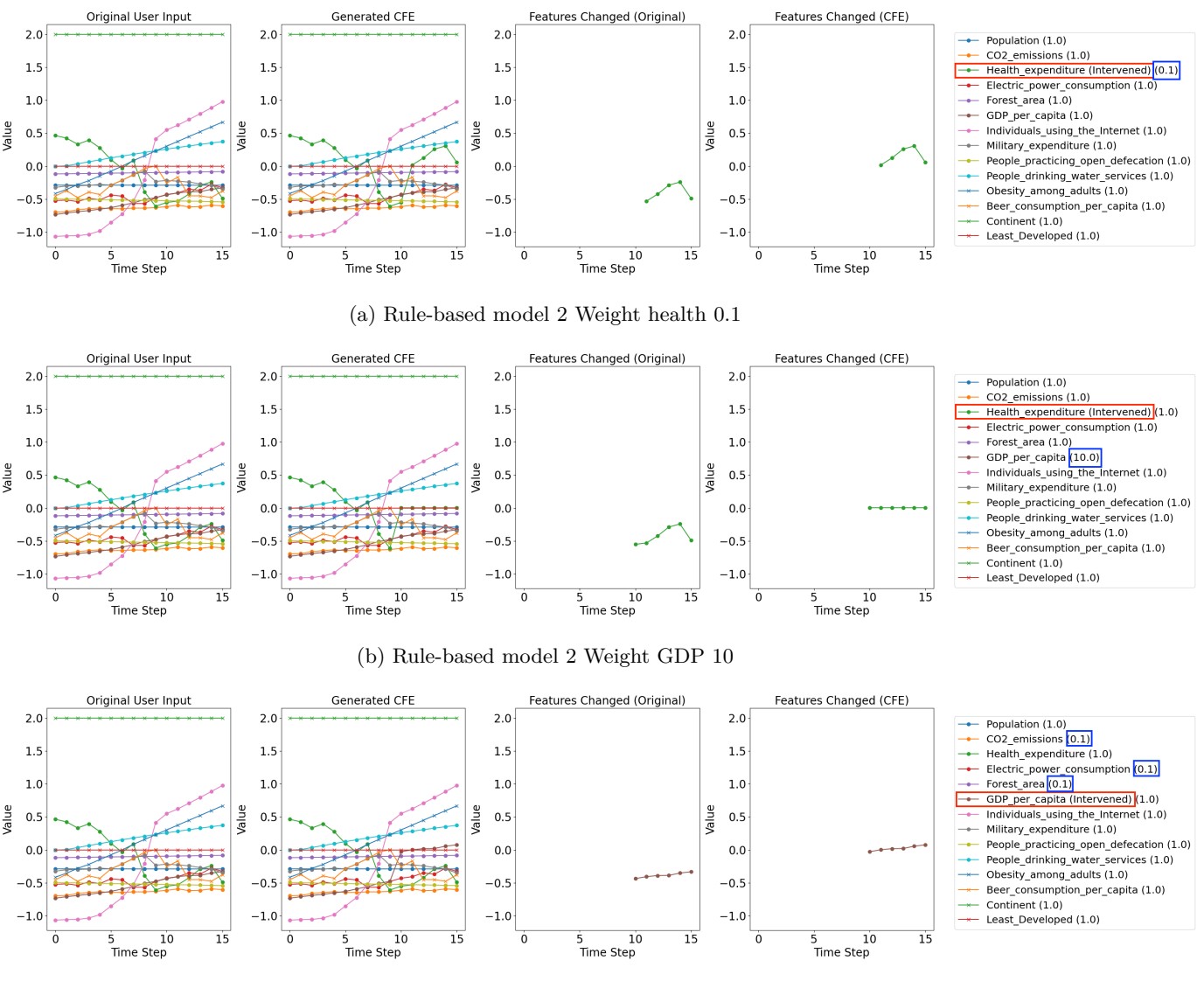

(a) Rule-based model 2 Weight health 0.1

(b) Rule-based model 2 Weight GDP 10

(c) Rule-based model 2 Weight CO2 Elec Forest 0.1

Figure 2: Qualitative examples with rule-based model 2 and different feature feasibility weights $W_{fsib}$.

and the second rule-based model as:

$$f(\boldsymbol{x}) = \begin{cases} Y' & \text{if } x^{\{k,d_1\}} > 0 \lor x^{\{k,d_2\}} > 0 \lor x^{\{k,d_3\}} > 0 \text{ for } K - 4 \leq k \leq K \\ Y & \text{otherwise} \end{cases}$$

where $Y'$ is the target class and $Y$ is the undesired class.

**Rule-based models for the Racket Sports dataset**

Let $d_i$ denote the $i$-th feature. We define the first rule-based model as:

$$f(\boldsymbol{x}) = \begin{cases} Y' & \text{if } x^{\{k,d_1\}} > 0 \land x^{\{k,d_5\}} > 0 \text{ for } K - 4 \leq k \leq K \\ Y & \text{otherwise} \end{cases}$$

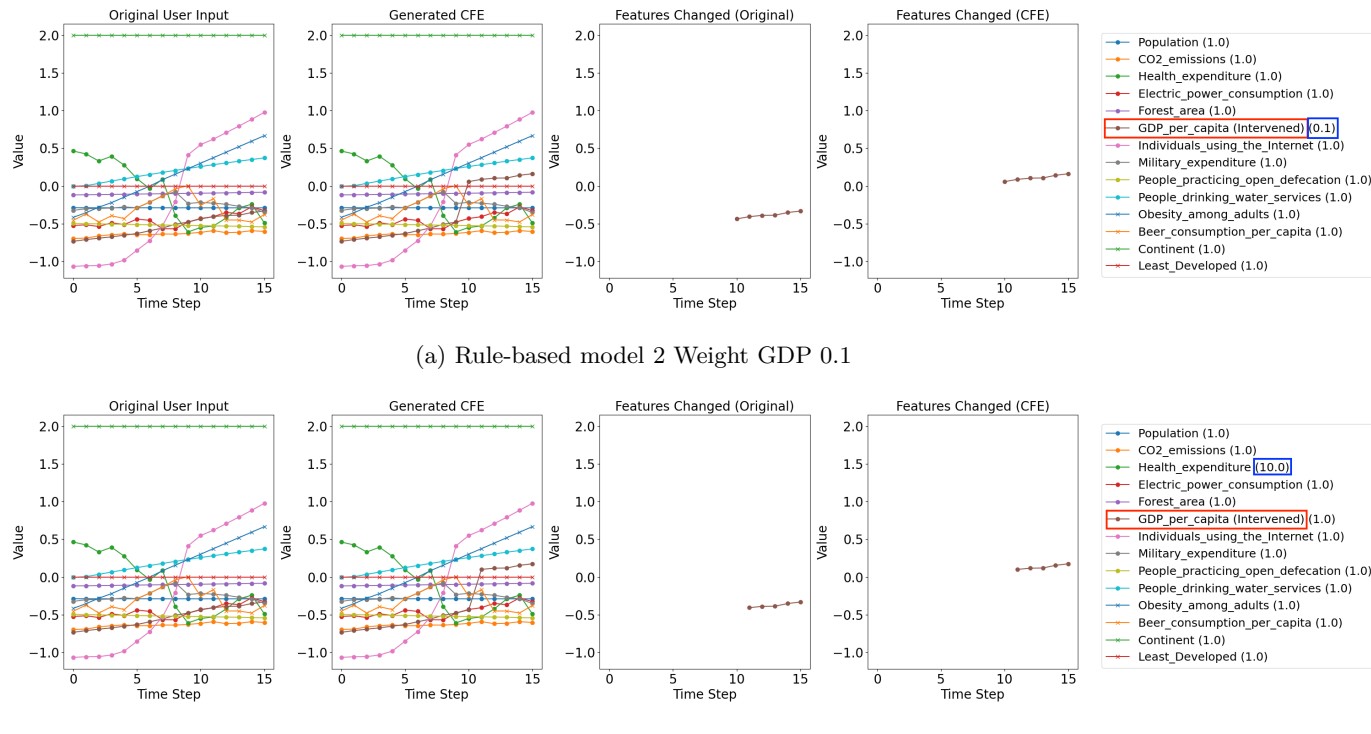

(a) Rule-based model 2 Weight GDP 0.1

(b) Rule-based model 2 Weight health 10

Figure 3: Qualitative examples with rule-based model 2 and different feature feasibility weights $W_{fsib}$. Among relevant features to the predictive model prediction, NTD-CFE prefers the features with smaller $W_{fsib}$.

and the second rule-based model as:

$$f(\boldsymbol{x}) = \begin{cases} Y' & \text{if } x^{\{k,d_1\}} > 0 \vee x^{\{k,d_5\}} > 0 \text{ for } K - 4 \leq k \leq K \\ Y & \text{otherwise} \end{cases}$$

where $Y'$ is the target class and $Y$ is the undesired class.

**Rule-based models for the Basic Motions dataset**

Let $d_i$ denote the $i$-th feature. We define the first rule-based model as:

$$f(\boldsymbol{x}) = \begin{cases} Y' & \text{if } x^{\{k,d_1\}} > 0 \wedge x^{\{k,d_3\}} > 0 \wedge x^{\{k,d_6\}} > 0 \text{ for } K - 9 \leq k \leq K \\ Y & \text{otherwise} \end{cases}$$

and the second rule-based model as:

$$f(\boldsymbol{x}) = \begin{cases} Y' & \text{if } x^{\{k,d_1\}} > 0 \vee x^{\{k,d_3\}} > 0 \vee x^{\{k,d_6\}} > 0 \text{ for } K - 9 \leq k \leq K \\ Y & \text{otherwise} \end{cases}$$

where $Y'$ is the target class "*Standing*", and $Y$ is the undesired class.

**Rule-based models for the eRing dataset**

Let $d_i$ denote the $i$-th feature. We define the first rule-based model as:

$$f(\boldsymbol{x}) = \begin{cases} Y' & \text{if } x^{\{k,d_2\}} > 0 \wedge x^{\{k,d_3\}} > 0 \text{ for } K - 9 \leq k \leq K \\ Y & \text{otherwise} \end{cases}$$

and the second rule-based model as:

$$f(\boldsymbol{x}) = \begin{cases} Y' & \text{if } x^{\{k,d_2\}} > 0 \vee x^{\{k,d_3\}} > 0 \text{ for } K - 9 \le k \le K \\ Y & \text{otherwise} \end{cases}$$

where $Y'$ is the target class and $Y$ is the undesired class.

**Rule-based models for the Japanese Vowels dataset**

Let $d_i$ denote the $i$-th feature. We define the first rule-based model as:

$$f(\boldsymbol{x}) = \begin{cases} Y' & \text{if } x^{\{k,d_1\}} > 0 \wedge x^{\{k,d_6\}} > 0 \wedge x^{\{k,d_12\}} > 0 \text{ for } K - 19 \le k \le K \\ Y & \text{otherwise} \end{cases}$$

and the second rule-based model as:

$$f(\boldsymbol{x}) = \begin{cases} Y' & \text{if } x^{\{k,d_1\}} > 0 \vee x^{\{k,d_6\}} > 0 \vee x^{\{k,d_12\}} > 0 \text{ for } K - 19 \le k \le K \\ Y & \text{otherwise} \end{cases}$$

where $Y'$ is the target class and $Y$ is the undesired class.

**Rule-based models for the Libras dataset**

Let $d_i$ denote the $i$-th feature. We define the first rule-based model as:

$$f(\boldsymbol{x}) = \begin{cases} Y' & \text{if } x^{\{k,d_1\}} > 0 \wedge x^{\{k,d_2\}} > 0 \text{ for } K - 19 \le k \le K \\ Y & \text{otherwise} \end{cases}$$

and the second rule-based model as:

$$f(\boldsymbol{x}) = \begin{cases} Y' & \text{if } x^{\{k,d_1\}} > 0 \vee x^{\{k,d_2\}} > 0 \text{ for } K - 19 \le k \le K \\ Y & \text{otherwise} \end{cases}$$

where $Y'$ is the target class and $Y$ is the undesired class.

## B.2 Other Predictive Models

Besides the rule-based models, each of the following predictive models is also used for each dataset.

**long short-term memory (LSTM).** The first layer of the neural network is a LSTM layer with 30 hidden states, followed by two linear layers. The first linear layer takes input of dimension of 30 and produces an output of dimension 60, then passes the output to a ReLU activation function. The second linear layer takes input of dimension of 60 and produces an output, then passes the output to a sigmoid activation function. We train the LSTM with learning rate 0.001 and weight decay 0.001 for 5000 epochs.

**K-nearest neighbor (KNN).** The number of neighbors to use for prediction is $\sqrt{N}$, where $N$ denotes the number of samples in the dataset.

**Random Forest.** The number of trees is 100. The minimum number of samples required to split an internal node is 2. The minimum number of samples required to be at a leaf node is 1.

## C Hyperparameters used in Section 5

We use a unique set of hyperparameter values for NTD-CFE throughout the paper, unless otherwise stated, without fine-tuning them:

- proximity weight $\lambda_{pxmt} = 0.001$

- maximum number of interventions per episode $M_T = 100$

- maximum number of episodes $M_E = 100$

- discount factor $\gamma = 0.99$

- learning rate $\alpha = 0.0001$

- regularization weight $\lambda_{WD} = 0.0$

The RL policy network contains two hidden linear layers with 1000 and 100 neurons, respectively. Adam (Kingma & Ba, 2014) is used as the optimizer.

It is important to note that NTD-CFE is not a supervised learning algorithm. As a reinforcement learning method, NTD-CFE does not rely on training datasets. Instead, it interacts with the environment (i.e., the predictive model $f$) for optimization. We select the set of hyperparameter values that works best for the Life Expectancy dataset from ten candidate sets of hyperparameter values. Although a more exhaustive hyperparameter search could potentially find another set of hyperparameters that produces better results, we leave this for future work. For the baseline methods, we use the hyperparameter values and architectures that are provided as default in their code [10]. All the experiments are conducted on CPU and with 32GB of RAM.

## D  Quantitative Experiments: NTD-CFE Versus NTD-CFE$_{in\_dist}$

In this section, we compare NTD-CFE, which does not enforce plausibility, with NTD-CFE$_{in\_dist}$, which enforces plausibility. For NTD-CFE$_{in\_dist}$, $F_{in\_dist}(\boldsymbol{x}) = \text{True}$ is applied (Line 17 of Algorithm 1) while any other hyperparameter values remain unchanged. We employ local outlier factor (LOF) (Breunig et al., 2000) as the (optional) oracular in-distribution detector $F_{in\_dist}$. LOF is a common method for assessing plausibility in CFE literature (Kanamori et al., 2020; Delaney et al., 2021; Wang et al., 2021b; Romashov et al., 2022). It employs KNNs to measure the degree to which a data point is unusual compared to others.

As shown in Tables 11 to 18, NTD-CFE$_{in\_dist}$ achieves plausibility rates of 100%. However, its success rates are lower than those of NTD-CFE in 22/39 cases. We further compare their proximity and sparsity under the same success rates. Although NTD-CFE$_{in\_dist}$ gets higher proximity in 8/17 cases and higher sparsity in 7/17 cases, the changes in the values are small.

---

[10]The code is publicly available at:

CoMTE: https://github.com/peaclab/CoMTE

Native-Guide: https://github.com/e-delaney/Instance-Based_CFE_TSC

CFRL: https://docs.seldon.io/projects/alibi/en/stable/methods/CFRL.html

FastAR: https://github.com/vsahil/FastAR-RL-for-generating-AR

Table 2: Features in the Life Expectancy dataset.

| Features | Type |
|---|---|
| *Country Name* | Categorical |
| *Year* | Categorical |
| *Continent* | Categorical |
| *Least Developed* | Categorical |
| *Population* | Continuous |
| *CO2 Emissions* | Continuous |
| *Health Expenditure* | Continuous |
| *Electric Power Consumption* | Continuous |
| *Forest Area* | Continuous |
| *GDP per Capita* | Continuous |
| *Individuals Using the Internet* | Continuous |
| *Military Expenditure* | Continuous |
| *People Practicing Open Defecation* | Continuous |
| *People Using at Least Basic Drinking Water Services* | Continuous |
| *Obesity Among Adults* | Continuous |
| *Beer Consumption per Capita* | Continuous |

**Label:** *Life Expectancy*  Categorical

Table 3: Quantitative results with the Life Expectancy dataset.

| Predictive Model | $N_{inv}$ | Methods | Success Rate (%) | Validity Rate (%) | Plausibility Rate (%) | Proximity | Sparsity |
|---|---|---|---|---|---|---|---|
| LSTM | 63 | CoMTE | **100** | **100** | **100** | 37.475 | 201.54 |
| | | Native-Guide | **100** | **100** | 85.714 | **30.674** | **190.238** |
| | | CFRL | 0 | — | — | — | — |
| | | FastAR | 1.587 | 1.587 | 100 | 0.45 | 1.0 |
| | | **NTD-CFE** | 98.413 | **100** | 80.645 | 10.893 | 64.065 |
| KNN | 68 | CoMTE | **100** | **100** | **100** | 46.366 | 204.176 |
| | | Native-Guide | **100** | **100** | 48.529 | $1.452 \times 10^{12}$ | **200.574** |
| | | CFRL | **100** | **100** | **100** | **44.264** | 206.824 |
| | | FastAR | 0 | — | — | — | — |
| | | **NTD-CFE** | 85.294 | **100** | 58.621 | 19.756 | 82.879 |
| Random Forest | 62 | CoMTE | **100** | **100** | **100** | 33.752 | 199.468 |
| | | Native-Guide | **100** | **100** | 79.032 | $2.834 \times 10^{13}$ | 200.548 |
| | | CFRL | **100** | **100** | **100** | 44.684 | 207.484 |
| | | FastAR | 0 | — | — | — | — |
| | | **NTD-CFE** | **100** | **100** | 96.774 | **8.724** | **49.661** |
| Rule-Based 1 | 87 | CoMTE | **100** | **100** | **100** | **47.542** | **203.747** |
| | | Native-Guide | 65.517 | 65.517 | 66.667 | $8.580 \times 10^{13}$ | 193.526 |
| | | CFRL | 0 | — | — | — | — |
| | | FastAR | 0 | — | — | — | — |
| | | **NTD-CFE** | 82.759 | **100** | 88.889 | 10.566 | 49.25 |
| Rule-Based 2 | 55 | CoMTE | **100** | **100** | **100** | **47.108** | **203.327** |
| | | Native-Guide | 72.727 | 72.727 | 60.0 | $7.749 \times 10^{13}$ | 183.025 |
| | | CFRL | 0 | — | — | — | — |
| | | FastAR | 0 | — | — | — | — |
| | | **NTD-CFE** | 81.818 | **100** | 86.667 | 11.238 | 52.667 |

Table 4: Quantitative results with the PEMS-SF dataset. Both FastAR and Native-Guide fail to run on this high-dimensional dataset, crashing due to memory issues, even after their memory allocation was increased to eight times that of the other methods.

| Predictive Model | $N_{inv}$ | Methods | Success Rate (%) | Validity Rate (%) | Plausibility Rate (%) | Proximity | Sparsity |
|---|---|---|---|---|---|---|---|
| LSTM | 64 | CoMTE | **100** | **100** | **81.25** | 139133.019 | **138672.0** |
| | | Native-Guide | — | — | — | — | — |
| | | CFRL | **100** | **100** | 0 | **139107.344** | **138672.0** |
| | | FastAR | — | — | — | — | — |
| | | **NTD-CFE** | 40.625 | **100** | 80.769 | 1511.488 | 961.654 |
| KNN | 96 | CoMTE | **100** | **100** | **74.118** | 139101.786 | **138672.0** |
| | | Native-Guide | — | — | — | — | — |
| | | CFRL | **100** | **100** | 0 | **139100.448** | **138672.0** |
| | | FastAR | — | — | — | — | — |
| | | **NTD-CFE** | 5.208 | **100** | 60.0 | 1657.268 | 1267.8 |
| Random Forest | 81 | CoMTE | **100** | **100** | **80.328** | 139134.124 | **138672.0** |
| | | Native-Guide | — | — | — | — | — |
| | | CFRL | **100** | **100** | 0 | **139105.953** | **138672.0** |
| | | FastAR | — | — | — | — | — |
| | | **NTD-CFE** | 0 | — | — | — | — |
| Rule Based 1 | 139 | CoMTE | **100** | **100** | **92.806** | **139122.140** | **138671.986** |
| | | Native-Guide | — | — | — | — | — |
| | | CFRL | 0 | — | — | — | — |
| | | FastAR | — | — | — | — | — |
| | | **NTD-CFE** | 96.403 | **100** | 73.881 | 3695.285 | 3520.687 |
| Rule Based 2 | 24 | CoMTE | **100** | **100** | **87.5** | 139067.896 | 138672.0 |
| | | Native-Guide | — | — | — | — | — |
| | | CFRL | 0 | — | — | — | — |
| | | FastAR | — | — | — | — | — |
| | | **NTD-CFE** | 100 | **100** | 58.333 | **2586.612** | **2357.750** |

Table 5: Quantitative results with the NATOPS dataset.

| Predictive Model | $N_{inv}$ | Methods | Success Rate (%) | Validity Rate (%) | Plausibility Rate (%) | Proximity | Sparsity |
|---|---|---|---|---|---|---|---|
| LSTM | 90 | CoMTE | **100** | **100** | **100** | 1285.262 | 1224.0 |
| | | Native-Guide | **100** | **100** | 63.333 | $1.270 \times 10^{13}$ | 1158.133 |
| | | CFRL | 0 | — | — | — | — |
| | | FastAR | 0 | — | — | — | — |
| | | **NTD-CFE** | **100** | **100** | 28.889 | **227.184** | **135.1** |
| KNN | 93 | CoMTE | **100** | **100** | **100** | **1284.259** | **1224.0** |
| | | Native-Guide | **100** | **100** | 55.914 | $6.332 \times 10^{13}$ | 1213.172 |
| | | CFRL | 0 | — | — | — | — |
| | | FastAR | 0 | — | — | — | — |
| | | **NTD-CFE** | 6.452 | **100** | 50.0 | 588.817 | 496.333 |
| Random Forest | 90 | CoMTE | **100** | **100** | **100** | 1285.888 | 1224.0 |
| | | Native-Guide | **100** | **100** | 28.889 | $3.193 \times 10^{12}$ | 927.722 |
| | | CFRL | **100** | **100** | 0 | 1277.502 | 1224.0 |
| | | FastAR | 0 | — | — | — | — |
| | | **NTD-CFE** | **100** | **100** | 45.556 | **228.323** | **157.733** |
| Rule-Based 1 | 178 | CoMTE | 0 | — | — | — | — |
| | | Native-Guide | 66.854 | 66.854 | 17.647 | $1.349 \times 10^{14}$ | 1207.563 |
| | | CFRL | 0 | — | — | — | — |
| | | FastAR | 0 | — | — | — | — |
| | | **NTD-CFE** | **96.629** | **100** | 86.047 | 188.263 | 144.105 |
| Rule-Based 2 | 126 | CoMTE | **100** | **100** | **100** | 1294.181 | 1224.0 |
| | | Native-Guide | 93.651 | 93.651 | 72.881 | $1.445 \times 10^{14}$ | 1208.458 |
| | | CFRL | 0 | — | — | — | — |
| | | FastAR | 0 | — | — | — | — |
| | | **NTD-CFE** | **100** | **100** | **100** | **33.756** | **20.587** |

Table 6: Quantitative results with the Heartbeat dataset.

| Predictive Model | $N_{inv}$ | Methods | Success Rate (%) | Validity Rate (%) | Plausibility Rate (%) | Proximity | Sparsity |
|---|---|---|---|---|---|---|---|
| LSTM | 136 | CoMTE | **100** | **100** | **100** | **621.692** | 609.926 |
| | | Native-Guide | **100** | **100** | 77.206 | $1.075 \times 10^{11}$ | **591.346** |
| | | CFRL | 2.941 | 2.941 | 100 | 627.946 | 610.0 |
| | | FastAR | 0 | — | — | — | — |
| | | **NTD-CFE** | 97.794 | **100** | 88.722 | 16.825 | 12.12 |
| KNN | 192 | CoMTE | **100** | **100** | **100** | **626.788** | **609.948** |
| | | Native-Guide | 97.396 | 97.396 | 60.963 | $4.749 \times 10^{12}$ | 600.824 |
| | | CFRL | 0 | — | — | — | — |
| | | FastAR | 0 | — | — | — | — |
| | | **NTD-CFE** | 72.396 | **100** | 30.935 | 145.611 | 132.288 |
| Random Forest | 147 | CoMTE | **100** | **100** | **100** | **622.636** | **609.864** |
| | | Native-Guide | 65.986 | 65.986 | 79.381 | $1.877 \times 10^{12}$ | 600.68 |
| | | CFRL | 0 | — | — | — | — |
| | | FastAR | 0 | — | — | — | — |
| | | **NTD-CFE** | 0.68 | **100** | 0 | 57.664 | 48.0 |
| Rule Based 1 | 171 | CoMTE | **100** | **100** | **100** | **624.442** | **609.883** |
| | | Native-Guide | 99.415 | 99.415 | 78.235 | $5.192 \times 10^{12}$ | 571.535 |
| | | CFRL | 0 | — | — | — | — |
| | | FastAR | 0 | — | — | — | — |
| | | **NTD-CFE** | 70.175 | **100** | 43.333 | 173.931 | 162.692 |
| Rule Based 2 | 120 | CoMTE | **100** | **100** | **100** | 619.454 | 609.917 |
| | | Native-Guide | **100** | **100** | 82.5 | $2.236 \times 10^{11}$ | 589.658 |
| | | CFRL | 0 | — | — | — | — |
| | | FastAR | 0 | — | — | — | — |
| | | **NTD-CFE** | **100** | **100** | 90.833 | **13.842** | **9.008** |

Table 7: Quantitative results with the Racket Sports dataset.

| Predictive Model | $N_{inv}$ | Methods | Success Rate (%) | Validity Rate (%) | Plausibility Rate (%) | Proximity | Sparsity |
|---|---|---|---|---|---|---|---|
| LSTM | 78 | CoMTE | **100** | **100** | **100** | 214.707 | 180.0 |
| | | Native-Guide | **100** | **100** | 75.641 | 202.626 | 161.59 |
| | | CFRL | 0 | — | — | — | — |
| | | FastAR | 14.103 | 14.103 | 100 | 1.786 | 1.091 |
| | | **NTD-CFE** | **100** | **100** | 98.718 | **16.734** | **8.295** |
| KNN | 112 | CoMTE | **100** | **100** | **100** | 217.73 | 180.0 |
| | | Native-Guide | **100** | **100** | 76.786 | $5.320 \times 10^{12}$ | 172.723 |
| | | CFRL | 0 | — | — | — | — |
| | | FastAR | 0 | — | — | — | — |
| | | **NTD-CFE** | **100** | **100** | 66.964 | **54.974** | **29.366** |
| Random Forest | 82 | CoMTE | **100** | **100** | **100** | 214.105 | 180.0 |
| | | Native-Guide | 98.78 | 98.78 | 77.778 | $3.529 \times 10^{12}$ | 167.222 |
| | | CFRL | 0 | — | — | — | — |
| | | FastAR | 0 | — | — | — | — |
| | | **NTD-CFE** | **100** | **100** | 90.244 | **43.695** | **26.232** |
| Rule Based 1 | 111 | CoMTE | **100** | **100** | **100** | **222.762** | **180.0** |
| | | Native-Guide | 94.595 | 94.595 | 67.619 | $1.170 \times 10^{14}$ | 172.486 |
| | | CFRL | 0 | — | — | — | — |
| | | FastAR | 0 | — | — | — | — |
| | | **NTD-CFE** | 98.198 | **100** | 93.578 | 24.46 | 13.385 |
| Rule Based 2 | 16 | CoMTE | **100** | **100** | **100** | 220.552 | 180.0 |
| | | Native-Guide | **100** | **100** | 75.0 | 228.065 | 170.125 |
| | | CFRL | 0 | — | — | — | — |
| | | FastAR | 0 | — | — | — | — |
| | | **NTD-CFE** | **100** | **100** | **100** | **16.183** | **9.438** |

Table 8: Quantitative results with the Basic Motions dataset.

| Predictive Model | $N_{inv}$ | Methods | Success Rate (%) | Validity Rate (%) | Plausibility Rate (%) | Proximity | Sparsity |
|---|---|---|---|---|---|---|---|
| LSTM | 14 | CoMTE | **100** | **100** | **100** | 782.188 | 600.0 |
| | | Native-Guide | **100** | **100** | 85.714 | 699.261 | 527.429 |
| | | CFRL | **100** | **100** | **100** | 802.065 | 600.0 |
| | | FastAR | 7.143 | 7.143 | 100 | 2.85 | 1.0 |
| | | **NTD-CFE** | **100** | **100** | **100** | **87.828** | **38.429** |
| KNN | 19 | CoMTE | **100** | **100** | **100** | 761.556 | 600.0 |
| | | Native-Guide | **100** | **100** | 78.947 | 706.719 | 562.526 |
| | | CFRL | **100** | **100** | **100** | 795.277 | 600.0 |
| | | FastAR | 0 | — | — | — | — |
| | | **NTD-CFE** | **100** | **100** | 94.737 | **206.984** | **121.421** |
| Random Forest | 20 | CoMTE | **100** | **100** | **100** | 751.741 | 600.0 |
| | | Native-Guide | **100** | **100** | 80.0 | 721.01 | 574.9 |
| | | CFRL | **100** | **100** | **100** | 773.104 | 600.0 |
| | | FastAR | 0 | — | — | — | — |
| | | **NTD-CFE** | **100** | **100** | 50.0 | **305.677** | **182.25** |
| Rule Based 1 | 35 | CoMTE | **100** | **100** | **100** | **896.874** | **600.0** |
| | | Native-Guide | 97.143 | 97.143 | 88.235 | 804.033 | 584.971 |
| | | CFRL | 0 | — | — | — | — |
| | | FastAR | 0 | — | — | — | — |
| | | **NTD-CFE** | 97.143 | **100** | 73.529 | 133.66 | 80.559 |
| Rule Based 2 | 8 | CoMTE | **100** | **100** | **100** | 703.79 | 600.0 |
| | | Native-Guide | **100** | **100** | 75.0 | 687.975 | 562.0 |
| | | CFRL | 0 | — | — | — | — |
| | | FastAR | 0 | — | — | — | — |
| | | **NTD-CFE** | **100** | **100** | **100** | **44.01** | **19.25** |

Table 9: Quantitative results with the Japanese Vowels dataset.

| Predictive Model | $N_{inv}$ | Methods | Success Rate (%) | Validity Rate (%) | Plausibility Rate (%) | Proximity | Sparsity |
|---|---|---|---|---|---|---|---|
| LSTM | 121 | CoMTE | **100** | **100** | **100** | 330.771 | 300.0 |
| | | Native-Guide | **100** | **100** | 88.43 | $3.725 \times 10^{12}$ | 281.05 |
| | | CFRL | **100** | **100** | **100** | 335.712 | 300.0 |
| | | FastAR | 0 | — | — | — | — |
| | | **NTD-CFE** | **100** | **100** | 99.174 | **35.3** | **19.413** |
| KNN | 124 | CoMTE | **100** | **100** | **100** | 331.845 | 300.0 |
| | | Native-Guide | **100** | **100** | 85.484 | $3.574 \times 10^{12}$ | 283.419 |
| | | CFRL | **100** | **100** | **100** | 334.799 | 300.0 |
| | | FastAR | 1.613 | 1.613 | 100 | 1.6 | 1.0 |
| | | **NTD-CFE** | **100** | **100** | 71.774 | **104.147** | **62.073** |
| Random Forest | 120 | CoMTE | **100** | **100** | **100** | 331.467 | 300.0 |
| | | Native-Guide | **100** | **100** | 89.167 | $9.824 \times 10^{12}$ | 290.233 |
| | | CFRL | **100** | **100** | **100** | 334.707 | 300.0 |
| | | FastAR | 0 | — | — | — | — |
| | | **NTD-CFE** | **100** | **100** | 90.0 | **83.856** | **52.55** |
| Rule Based 1 | 269 | CoMTE | 0 | — | — | — | — |
| | | Native-Guide | 30.855 | 30.855 | 78.313 | $3.032 \times 10^{13}$ | 300.0 |
| | | CFRL | 0 | — | — | — | — |
| | | FastAR | 0 | — | — | — | — |
| | | **NTD-CFE** | **53.903** | **100** | 57.241 | 166.07 | 123.869 |
| Rule Based 2 | 149 | CoMTE | **100** | **100** | **100** | 330.597 | 300.0 |
| | | Native-Guide | 67.114 | 67.114 | 77.0 | $6.965 \times 10^{12}$ | 289.92 |
| | | CFRL | 0 | — | — | — | — |
| | | FastAR | 0 | — | — | — | — |
| | | **NTD-CFE** | **100** | **100** | 97.987 | **38.206** | **18.436** |

Table 10: Quantitative results with the Libras dataset.

| Predictive Model | $N_{inv}$ | Methods | Success Rate (%) | Validity Rate (%) | Plausibility Rate (%) | Proximity | Sparsity |
|---|---|---|---|---|---|---|---|
| LSTM | 89 | CoMTE | **100** | **100** | **100** | 120.329 | 89.494 |
| | | Native-Guide | **100** | **100** | 64.045 | 111.969 | 88.382 |
| | | CFRL | 0 | — | — | — | — |
| | | FastAR | 59.551 | 59.551 | 64.151 | 2.037 | 1.094 |
| | | **NTD-CFE** | **100** | **100** | 32.584 | **21.421** | **12.73** |
| KNN | 89 | CoMTE | **100** | **100** | **100** | 118.379 | 88.989 |
| | | Native-Guide | **100** | **100** | 51.685 | 126.318 | 89.213 |
| | | CFRL | **100** | **100** | **100** | 139.54 | 90.0 |
| | | FastAR | 1.124 | 1.124 | 0 | 4.4 | 3.0 |
| | | **NTD-CFE** | **100** | **100** | 14.607 | **45.306** | **24.112** |
| Random Forest | 84 | CoMTE | **100** | **100** | **100** | 108.074 | 88.393 |
| | | Native-Guide | **100** | **100** | 75.0 | 111.086 | 87.143 |
| | | CFRL | 0 | — | — | — | — |
| | | FastAR | 0 | — | — | — | — |
| | | **NTD-CFE** | **100** | **100** | 13.095 | **50.151** | **27.024** |
| Rule Based 1 | 116 | CoMTE | **100** | **100** | **100** | 144.053 | 88.836 |
| | | Native-Guide | **100** | **100** | 87.069 | 135.861 | 88.836 |
| | | CFRL | 0 | — | — | — | — |
| | | FastAR | 0 | — | — | — | — |
| | | **NTD-CFE** | **100** | **100** | 6.034 | **74.67** | **39.647** |
| Rule Based 2 | 53 | CoMTE | **100** | **100** | **100** | 131.461 | 90.0 |
| | | Native-Guide | **100** | **100** | 98.113 | 135.119 | 90.0 |
| | | CFRL | 0 | — | — | — | — |
| | | FastAR | 0 | — | — | — | — |
| | | **NTD-CFE** | **100** | **100** | 20.755 | **45.037** | **23.962** |

Table 11: Compare NTD-CFE and NTD-CFE$_{in\_dist}$ with the Life Expectancy dataset.

| Predictive Model | $N_{inv}$ | Methods | Success Rate (%) | Validity Rate (%) | Plausibility Rate (%) | Proximity | Sparsity |
|---|---|---|---|---|---|---|---|
| LSTM | 63 | NTD-CFE | 98.413 | 100 | 80.645 | 10.893 | 64.065 |
| | | NTD-CFE$_{in\_dist}$ | 93.651 | 100 | 100 | 12.546 | 57.695 |
| KNN | 68 | NTD-CFE | 85.294 | 100 | 58.621 | 19.756 | 82.879 |
| | | NTD-CFE$_{in\_dist}$ | 79.412 | 100 | 100 | 25.653 | 82.241 |
| Random Forest | 62 | NTD-CFE | 100 | 100 | 96.774 | 8.724 | 49.661 |
| | | NTD-CFE$_{in\_dist}$ | 100 | 100 | 100 | 8.984 | 49.242 |
| Rule Based 1 | 87 | NTD-CFE | 82.759 | 100 | 88.889 | 10.566 | 49.25 |
| | | NTD-CFE$_{in\_dist}$ | 79.31 | 100 | 100 | 10.063 | 46.014 |
| Rule Based 2 | 55 | NTD-CFE | 81.818 | 100 | 86.667 | 11.238 | 52.667 |
| | | NTD-CFE$_{in\_dist}$ | 76.364 | 100 | 100 | 9.329 | 46.69 |

Table 12: Compare NTD-CFE and NTD-CFE$_{in\_dist}$ with the NATOPS dataset.

| Predictive Model | $N_{inv}$ | Methods | Success Rate (%) | Validity Rate (%) | Plausibility Rate (%) | Proximity | Sparsity |
|---|---|---|---|---|---|---|---|
| LSTM | 90 | NTD-CFE | 100 | 100 | 28.889 | 227.184 | 135.1 |
| | | NTD-CFE$_{in\_dist}$ | 40.0 | 100 | 100 | 192.739 | 125.5 |
| KNN | 93 | NTD-CFE | 6.452 | 100 | 50.0 | 588.817 | 496.333 |
| | | NTD-CFE$_{in\_dist}$ | 3.226 | 100 | 100 | 109.59 | 67.0 |
| Random Forest | 90 | NTD-CFE | 100 | 100 | 45.556 | 228.323 | 157.733 |
| | | NTD-CFE$_{in\_dist}$ | 63.333 | 100 | 100 | 212.14 | 156.333 |
| Rule Based 1 | 178 | NTD-CFE | 96.629 | 100 | 86.047 | 188.263 | 144.105 |
| | | NTD-CFE$_{in\_dist}$ | 90.449 | 100 | 100 | 133.034 | 95.646 |
| Rule Based 2 | 126 | NTD-CFE | 100 | 100 | 100 | 33.756 | 20.587 |
| | | NTD-CFE$_{in\_dist}$ | 100 | 100 | 100 | 33.756 | 20.587 |

Table 13: Compare NTD-CFE and NTD-CFE$_{in\_dist}$ with the Heartbeat dataset.

| Predictive Model | $N_{inv}$ | Methods | Success Rate (%) | Validity Rate (%) | Plausibility Rate (%) | Proximity | Sparsity |
|---|---|---|---|---|---|---|---|
| LSTM | 136 | NTD-CFE | 97.794 | 100 | 88.722 | 16.825 | 12.12 |
| | | NTD-CFE$_{in\_dist}$ | 97.794 | 100 | 100 | 19.104 | 14.714 |
| KNN | 192 | NTD-CFE | 72.396 | 100 | 30.935 | 145.611 | 132.288 |
| | | NTD-CFE$_{in\_dist}$ | 25.521 | 100 | 100 | 87.665 | 77.245 |
| Random Forest | 147 | NTD-CFE | 0.68 | 100 | 0 | 57.664 | 48.0 |
| | | NTD-CFE$_{in\_dist}$ | 0 | — | — | — | — |
| Rule Based 1 | 171 | NTD-CFE | 70.175 | 100 | 43.333 | 173.931 | 162.692 |
| | | NTD-CFE$_{in\_dist}$ | 30.994 | 100 | 100 | 83.472 | 75.906 |
| Rule Based 2 | 120 | NTD-CFE | 100 | 100 | 90.833 | 13.842 | 9.008 |
| | | NTD-CFE$_{in\_dist}$ | 99.167 | 100 | 100 | 14.205 | 9.613 |

Table 14: Compare NTD-CFE and NTD-CFE$_{in\_dist}$ with the Racket Sports dataset.

| Predictive Model | $N_{inv}$ | Methods | Success Rate (%) | Validity Rate (%) | Plausibility Rate (%) | Proximity | Sparsity |
|---|---|---|---|---|---|---|---|
| LSTM | 78 | NTD-CFE | 100 | 100 | 98.718 | 16.734 | 8.295 |
| | | NTD-CFE$_{in\_dist}$ | 100 | 100 | 100 | 16.961 | 8.423 |
| KNN | 112 | NTD-CFE | 100 | 100 | 66.964 | 54.974 | 29.366 |
| | | NTD-CFE$_{in\_dist}$ | 100 | 100 | 100 | 64.028 | 36.42 |
| Random Forest | 82 | NTD-CFE | 100 | 100 | 90.244 | 43.695 | 26.232 |
| | | NTD-CFE$_{in\_dist}$ | 100 | 100 | 100 | 44.611 | 27.768 |
| Rule Based 1 | 111 | NTD-CFE | 98.198 | 100 | 93.578 | 24.46 | 13.385 |
| | | NTD-CFE$_{in\_dist}$ | 98.198 | 100 | 100 | 25.226 | 13.633 |
| Rule Based 2 | 16 | NTD-CFE | 100 | 100 | 100 | 16.183 | 9.438 |
| | | NTD-CFE$_{in\_dist}$ | 100 | 100 | 100 | 16.183 | 9.438 |

Table 15: Compare NTD-CFE and NTD-CFE$_{in\_dist}$ with the Basic Motions dataset.

| Predictive Model | $N_{inv}$ | Methods | Success Rate (%) | Validity Rate (%) | Plausibility Rate (%) | Proximity | Sparsity |
|---|---|---|---|---|---|---|---|
| LSTM | 14 | NTD-CFE | 100 | 100 | 100 | 87.828 | 38.429 |
| | | NTD-CFE$_{in\_dist}$ | 100 | 100 | 100 | 87.828 | 38.429 |
| KNN | 19 | NTD-CFE | 100 | 100 | 94.737 | 206.984 | 121.421 |
| | | NTD-CFE$_{in\_dist}$ | 100 | 100 | 100 | 223.681 | 128.368 |
| Random Forest | 20 | NTD-CFE | 100 | 100 | 50.0 | 305.677 | 182.25 |
| | | NTD-CFE$_{in\_dist}$ | 95.0 | 100 | 100 | 321.474 | 199.895 |
| Rule Based 1 | 35 | NTD-CFE | 97.143 | 100 | 73.529 | 133.66 | 80.559 |
| | | NTD-CFE$_{in\_dist}$ | 97.143 | 100 | 100 | 157.419 | 104.824 |
| Rule Based 2 | 8 | NTD-CFE | 100 | 100 | 100 | 44.01 | 19.25 |
| | | NTD-CFE$_{in\_dist}$ | 100 | 100 | 100 | 44.01 | 19.25 |

Table 16: Compare NTD-CFE and NTD-CFE$_{in\_dist}$ with the eRing dataset.

| Predictive Model | $N_{inv}$ | Methods | Success Rate (%) | Validity Rate (%) | Plausibility Rate (%) | Proximity | Sparsity |
|---|---|---|---|---|---|---|---|
| LSTM | 14 | NTD-CFE | 100 | 100 | 100 | 54.682 | 31.214 |
| | | NTD-CFE$_{in\_dist}$ | 100 | 100 | 100 | 54.682 | 31.214 |
| KNN | 16 | NTD-CFE | 100 | 100 | 100 | 144.937 | 86.688 |
| | | NTD-CFE$_{in\_dist}$ | 100 | 100 | 100 | 144.937 | 86.688 |
| Random Forest | 15 | NTD-CFE | 100 | 100 | 100 | 68.882 | 46.733 |
| | | NTD-CFE$_{in\_dist}$ | 100 | 100 | 100 | 68.882 | 46.733 |
| Rule Based 1 | 29 | NTD-CFE | 100 | 100 | 100 | 103.953 | 63.345 |
| | | NTD-CFE$_{in\_dist}$ | 100 | 100 | 100 | 103.953 | 63.345 |
| Rule Based 2 | 25 | NTD-CFE | 100 | 100 | 100 | 31.301 | 14.76 |
| | | NTD-CFE$_{in\_dist}$ | 100 | 100 | 100 | 31.301 | 14.76 |

Table 17: Compare NTD-CFE and NTD-CFE$_{in\_dist}$ with the Japanese Vowels dataset.

| Predictive Model | $N_{inv}$ | Methods | Success Rate (%) | Validity Rate (%) | Plausibility Rate (%) | Proximity | Sparsity |
|---|---|---|---|---|---|---|---|
| LSTM | 121 | NTD-CFE | 100 | 100 | 99.174 | 35.3 | 19.413 |
| | | NTD-CFE$_{in\_dist}$ | 99.174 | 100 | 100 | 35.753 | 19.833 |
| KNN | 124 | NTD-CFE | 100 | 100 | 71.774 | 104.147 | 62.073 |
| | | NTD-CFE$_{in\_dist}$ | 94.355 | 100 | 100 | 114.437 | 74.667 |
| Random Forest | 120 | NTD-CFE | 100 | 100 | 90.0 | 83.856 | 52.55 |
| | | NTD-CFE$_{in\_dist}$ | 98.333 | 100 | 100 | 85.103 | 54.314 |
| Rule Based 1 | 269 | NTD-CFE | 53.903 | 100 | 57.241 | 166.07 | 123.869 |
| | | NTD-CFE$_{in\_dist}$ | 34.201 | 100 | 100 | 135.475 | 98.674 |
| Rule Based 2 | 149 | NTD-CFE | 100 | 100 | 97.987 | 38.206 | 18.436 |
| | | NTD-CFE$_{in\_dist}$ | 99.329 | 100 | 100 | 38.038 | 18.926 |

Table 18: Compare NTD-CFE and NTD-CFE$_{in\_dist}$ with the Libras dataset.

| Predictive Model | $N_{inv}$ | Methods | Success Rate (%) | Validity Rate (%) | Plausibility Rate (%) | Proximity | Sparsity |
|---|---|---|---|---|---|---|---|
| LSTM | 89 | NTD-CFE | 100 | 100 | 32.584 | 21.421 | 12.73 |
| | | NTD-CFE$_{in\_dist}$ | 93.258 | 100 | 100 | 39.937 | 23.831 |
| KNN | 89 | NTD-CFE | 100 | 100 | 14.607 | 45.306 | 24.112 |
| | | NTD-CFE$_{in\_dist}$ | 88.764 | 100 | 100 | 63.915 | 37.215 |
| Random Forest | 84 | NTD-CFE | 100 | 100 | 13.095 | 50.151 | 27.024 |
| | | NTD-CFE$_{in\_dist}$ | 88.095 | 100 | 100 | 61.098 | 40.635 |
| Rule Based 1 | 116 | NTD-CFE | 100 | 100 | 6.034 | 74.67 | 39.647 |
| | | NTD-CFE$_{in\_dist}$ | 50.862 | 100 | 100 | 87.461 | 50.695 |
| Rule Based 2 | 53 | NTD-CFE | 100 | 100 | 20.755 | 45.037 | 23.962 |
| | | NTD-CFE$_{in\_dist}$ | 84.906 | 100 | 100 | 75.344 | 38.289 |

Table 19: This table shows that success rates are improved with increased maximum number of episodes $M_E$ and maximum number of interventions per episode $M_T$, which correspond to more exhaustive RL search. We only test against the cases from Tables 1 and 3 to 10 in which the default hyperparameters give success rates below 50%.

| NTD-CFE Hyperparameters | Success Rate (%) | Validity Rate (%) | Plausibility Rate (%) | Proximity | Sparsity |
|---|---|---|---|---|---|
| $M_E = 100, M_T = 100$ | 0.68 | 100 | 0 | 57.664 | 48.0 |
| $M_E = 1000, M_T = 100$ | 10.204 | 100 | 20.0 | 267.045 | 253.2 |
| $M_E = 1000, M_T = 1000$ | 76.87 | 100 | 4.425 | 444.993 | 417.336 |

(a) Dataset: Heartbeat. Predictive model: random forest.

| NTD-CFE Hyperparameters | Success Rate (%) | Validity Rate (%) | Plausibility Rate (%) | Proximity | Sparsity |
|---|---|---|---|---|---|
| $M_E = 100, M_T = 100$ | 6.452 | 100 | 50.0 | 588.817 | 496.333 |
| $M_E = 1000, M_T = 100$ | 12.903 | 100 | 33.333 | 711.761 | 595.583 |
| $M_E = 10000, M_T = 100$ | 26.882 | 100 | 16.0 | 782.599 | 627.96 |

(b) Dataset: NATOPS. Predictive model: KNN.

