# OpenReview forum: "No $D_{train}$: Model-Agnostic Counterfactual Explanations Using Reinforcement Learning"
_TMLR — Accepted by TMLR_

### Review · Reviewer_U7fP · 2025-01-21

**Summary Of Contributions:**

The authors propose *NTD-CFE*, a model-agnostic counterfactual explanation method for low-dimensional time series data. *NTD-CFE* is based on reinforcement learning and does not require training data (except for rescaling features), in constrast to similar existing methods. The method is experimentally demonstrated on multiple real-world data sets and compared against other counterfactual explanation methods that are based on reinforcement learning.

**Audience:**

Yes

**Broader Impact Concerns:**

None.

**Claims And Evidence:**

No

**Requested Changes:**

I will proceed to list requested changes one-by-one. Points that are critical for meeting the acceptance criteria are highlighted by *(critical)*.

**Abstract**

- *When ML methods are responsible for making critical decisions, stakeholders often
require insights into how to alter these decisions* I am not sure if this is a good way of putting it. How about *When ML methods are responsible for making critical decisions, stakeholders often
require and understanding of how this decision came about*?

- *most existing CFE methods require access to the model’s training dataset, few methods can handle multivariate time-series, and none of model-agnostic CFE methods can handle multivariate time-series without training datasets*. First of all, I believe italics should be used a bit more sparingly. Also, I disagree that training data is required for computing CFE. From what I understand, the only thing that is required is a machine learning model that has been trained and a factual example. Please clarify.

**Introduction**

- *Counterfactual Explanations (CFEs) (Wachter et al., 2017) were introduced to fill this gap* I would disagree with this statement. A counterfactual explanation only tells what must have been different for a different prediction to be made. Actionability in counterfactual explanations has been critized and discussed by many authors [1, 2, 3]. Please clarify.

- *For instance, a CFE that shows if Bob’s name were Alice* This is actually a perfect example regarding my previous point. Changing one's name is not actionable and vanilla counterfactual explanations generate such outputs.

- *(critical)* *Additionally, to the best of our knowledge, all existing model-agnostic CFE methods for multivariate time-series require access to large collection of samples from the training distribution of the model being explained. This requirement can be infeasible in real-world domains especially due to privacy and other concerns.* Given that not requiring training data seems to lie at the core of the manuscript, I am surprised to find no information about how and why these existing methods require a training set. I believe there should be a major section that discusses this aspect.

- *high dimensionality* I cannot find experiments with high-dimensional data in the experiments. Please remove the claim or demonstrate experiments on high-dimensional data sets for time series such as [4].

**4 The proposed method: NTD-CFE**

- *from $\mathbf{x}^*$ to $\mathbf{O}^*$*. I find this notation a bit strange. Why not simply call the factual point $\mathbf{x}$ and the counterfactual $\mathbf{x}'$ or similar? Please consider changing this, unless there is a good reason to have the original.

- *We assume that the prediction function of f computes quickly* Please consider replacing by something like *We assume that the prediction function of f is computationally efficient*.

- *can optionally be divided into actionable features $\mathbf{D}_{\text{act}}$, which...* I would assume that these sets can be overlapping (?). If yes, then the word *divided* does not seem correct. If they cannot be overlapping, why not? Please clarify.

- *(critical)* **State Transition** I am greatly struggling to understand the relevance of this state transition component, which is also related to weakness 1. What would happen if one were to remove this component and instead have the policy directly output the next state $x_{t+1}$? It would be important to showcase an ablation that highlights the relevance of this component. Additional explanation in the main text would also be relevant.

- *Reward and Proximity* This choice of reward function seems quite inefficient, because the agent needs to learn the space of feasible solutions itself. In addition, It cannot be that this reward function effectively leads to solutions that are valid. I guess the authors consider some sort of epsilon radius around the counterfactual value to be valid. I believe that it would be much more effective to address this circumstance with constrained optimization methods such as the augmented Lagrangian method, which should also be applicable to zero-order optimization. Please discuss.

**Algorithm 1**

- I appreciate that the authors make use of different colors. However, the type of gray used seems a bit too light for my taste. Would it be possible to choose a slightly darker color for this?

- line 16: I find this line in the algorithm quite sketchy. Does that mean one needs to run the algorithm and hope that at some point a valid counterfactual is guessed? This sounds very inefficient, considering that there exist methods for constrained optimization that are guaranteed to yield valid solutions on the first try, like aforementioned augmented Lagrangian method (see weakness 3).

**5.2.1 Results**

*Comparison with CoMTE* The authors write that CoMTE requires training data. It would be very important to know how this method uses training data. It seems also not totally correct that the proposed method does not require any training data, as mentioned in *4.1 Limitations*. To evaluate the contribution, it would be important to know more about what these baselines do.

**Figure 1**

*(critical)* I do not understand this figure at all. For instance, why would the original user input change over time? And why does such a prominent figure showcase what rule-based models do, without even a comparison to the proposed algorithm? And what is even the counterfactual value in these plots? I believe this figure needs to be greatly revised or omitted.

[1] Karimi, Amir-Hossein, Bernhard Schölkopf, and Isabel Valera. "Algorithmic recourse: from counterfactual explanations to interventions." Proceedings of the 2021 ACM conference on fairness, accountability, and transparency. 2021.

[2] Karimi, Amir-Hossein, et al. "Algorithmic recourse under imperfect causal knowledge: a probabilistic approach." Advances in neural information processing systems 33 (2020): 265-277.

[3] Kladny, Klaus-Rudolf, et al. "Deep backtracking counterfactuals for causally compliant explanations." Transactions on Machine Learning Research (2024).

[4] Requena-Mesa, Christian, et al. "EarthNet2021: A large-scale dataset and challenge for Earth surface forecasting as a guided video prediction task." Proceedings of the IEEE/CVF Conference on Computer Vision and Pattern Recognition. 2021.

**Strengths And Weaknesses:**

**Strengths**

1. Overall, the manuscript is well-written and easy to follow.

2. The proposed algorithm is simple and straight-forward. In general, the manuscript should be accessible to a broad audience.

3. Many experiments on real-world data sets are conducted.

**Weaknesses**

Generally, I believe there is still a lot of work to do until this manuscript meets the acceptance criteria.

1. It is not clear at all what the relevance of reinforcement learning is to the proposed method. The authors rightfully criticize that most counterfactual explanation methods require differentiability of the used predictor, which can be restrictive. However, there exist many zero-order optimization methods like Bayesian optimization that seem more straightforward to me and do not require gradients either. The fundamental components to reinforcement learning are 1) state transition dynamics, 2) a stochastic environment, and 3) an expected cumulative reward. All of these components seem to be missing here. The soundness of using reinforcement learning for the purpose of generating counterfactual explanations needs to be greatly and critically discussed.

2. While I appreciate the comparison to many other RL based methods, the most relevant baselines are missing, in my opinion. This point is strongly related to weakness 1 and I believe it would be very important to demonstrate at least one more straight-forward approach to solving the problem, in comparison. Specifically, I recommend comparing to methods based on Bayesian optimization such as the one proposed by [1].

3. The proposed algorithm seems quite naive and I am convinced that this problem can be solved both more efficiently and effectively. For instance, I am highly sceptical about the choice of reward function which simply puts zero reward on infeasible solutions. There are many more effective ways of dealing with constraints in optimization that don't require the user to ``hope'' for valid counterfactuals, but ensure validity of counterfactuals by design of the optimization procedure (https://en.wikipedia.org/wiki/Constrained_optimization). I believe that such methods for taking into accounts should be greatly discussed as well.

[1] Spooner, Thomas, et al. "Counterfactual explanations for arbitrary regression models." arXiv preprint arXiv:2106.15212 (2021).

---

> ### Author Response · Authors · 2025-01-27
>
> Dear Reviewer U7fP,
>
> Thank you very much for reviewing our work and providing constructive feedback.
>
> Re: Relevance of reinforcement learning (RL) over Bayesian optimization (BO); inefficient compared to optimization methods
>
> > To solve the CFE problem without training dataset, we need some search algorithms. For multivariate time-series data, the alteration space (timestep × multivariate) to change the original $x^*$ to new CFEs is high and continuous. BO is usually not scalable. Existing BO-based CFE methods primarily focus on static or univariate time-series data and are unsuitable for multivariate time-series data, which is our focus. Instead, we formulate this search problem as a continuous control task using RL, reducing the control problem to a 3-dimensional space (as detailed in Section 4: "Each action $a$ ... is 3-dimensional ...").
> >
> >Additionally, causal constraints are essential for CFE methods [1]. While BO-based methods can impose value constraints (e.g., bounding ranges), encoding causality—such as how changes in one time step affect others—is more naturally handled in our RL framework.
> >
> > [1] Verma, Sahil, et al. "Counterfactual explanations and algorithmic recourses for machine learning: A review." _ACM Computing Surveys_ 56.12 (2024): 1-42.
>
>
> Re: State transition, stochastic environment and cumulative reward in RL are missing.
>
> >Section 4 and Algorithm 1 define all three components:
> >- **State transition**: Represents the path from the original input $x^*$ to the generated CFE.
> >- **Environment**: While stochasticity is not required, RL can operate in deterministic environments as well. In our case, the environment is the predictive model $f$.
> >- **Cumulative reward**: Encourages both the desired model prediction and proximity between $x^*$ and the generated CFEs.
>
>
> Re: Compare to baselines based on Bayesian optimization
>
> > We did not compare with BO-based baselines because we are not aware of BO-based CFE methods specifically designed for multivariate time-series data.
>
>
> Re: Disagreement on training data is required for computing CFEs; How and why existing methods require training datasets; important to know how CoMTE uses training data;
>
> > Although there are optimization-based CFE methods that do not require training datasets, other CFE methods do require training datasets. In Section 2 (Related Works), we explain how the baselines, including CoMTE, utilize training datasets.
>
>
> Re: High dimensionality
>
> > We did not claim that our method addresses "high-dimensional" time-series data. The only place we mentioned "high dimensionality" is "However, CFE methods for multivariate time-series data are less common due to the challenges posed by high dimensionality". Here, we compare multivariate time-series data to static data, which inherently has lower dimensionality. We acknowledge that "high-dimensionality" may be a misleading term and will revise the sentence for clarity.
>
>
> Re: Have the policy network directly output the next $x$
>
> > While having the policy network directly output the next $x$ could be feasible, we argue that formulating this task as an RL problem provides a more structured and effective approach than solving it generatively.
>
>
> Re: What Figure 1 demonstrates
>
> > Figure 1 illustrates the following (from left to right):
> > 1. The original feature values across time steps.
> > 2. The feature values across time steps of a CFE generated by the proposed algorithm.
> > 3. The specific parts of the original features altered by the proposed algorithm to transition from (1) to (2).
> > 4. The new values in the generated CFE.
>
>
> Thank you very much. We will add BO-based methods in related works and clarify the relevance of RL.

---

> > ### Comment · Reviewer_U7fP · 2025-03-15
> >
> > I thank the authors for their response. I am still sceptical whether the relevance of reinforcement learning is highlighted enough. In the revised version, the authors write that Bayesian optimization (BO) cannot be applied to temporal data. I would challenge this claim. Maybe the point that the authors want to make is that BO is often regarded to be inefficient for high-dimensional data (in principle, one could treat the entire time series as one single data point and perform BO on that). However, I have made personal experience that BO works surprisingly well in high dimensions. If the authors could be more precise about this highly relevant point, I will recommend acceptance.

---

> > > ### Author Response · Authors · 2025-03-16
> > >
> > > Thank you Reviewer U7fP for your follow-up. We found the discussion beneficial.
> > >
> > > We are only aware of BO-based CFE methods designed for static or univariate time-series data, whereas our work focuses on multivariate time-series data. These BO-based CFE methods do not account for complex causal constraints in CFEs, which we consider a significant limitation. For this reason, we believe RL is a more suitable approach.
> > >
> > > Regarding the complexity of applying BO to this problem, we do not assume how a predictive model makes predictions (e.g. which time steps in the data the predictive model depends on). Consequently, in a BO setting, each time step in a time-series can be treated as a separate hyperparameter. When dealing with multiple time-series, this leads to an excessive number of hyperparameters. For instance, in the Heartbeat dataset, there are 61 time-series, each with 405 time steps, resulting in a total of 61 x 405 = 24,705 hyperparameters. This appears to be an overwhelming number of hyperparameters for BO.

---

### Review · Reviewer_AiGr · 2025-02-01

**Summary Of Contributions:**

This work introduces a model-agnostic counterfactual explanation method for multivariate time-series data, leveraging reinforcement learning (RL) to identify key features for generating counterfactuals.

**Audience:**

Yes

**Claims And Evidence:**

No

**Requested Changes:**

Refer to the weaknesses section.

**Strengths And Weaknesses:**

Strengths:
    1. This study tackles an important problem—counterfactual explanations—and the integration of RL for feature selection is a promising approach.
    2. The method is validated on multiple real-world datasets, highlighting its practical applicability.


Weakness:
1. Novelty:
    Several RL-based counterfactual methods already exist. A key claimed contribution is that the proposed method does not require training data from the model being explained. However, this claim remains unclear, as the paper does not specify how the RL model is trained.

2. Presentation & Readability:
    1)	The motivation and core ideas of the proposed method should be introduced and summarized more clearly.
    2)	Algorithm 1 outlines the explanation generation process, but its description is too high-level and lacks essential details. For example:
        a. How is the RL model trained?
        b. How are constraints incorporated in line 13?
        c. Providing more details on these aspects would improve clarity.
3. Soundness:
    1) The paper does not explain how the learnable parameters of the RL model are obtained.
    2) The action space and feature selection process need further clarification:
        a. Are candidate features selected one-by-one or in batches?
        b. How is the policy model updated? Given the high-dimensional feature space, is the approach computationally scalable?

4. The reliability of the results is unclear:
    a. How are the counterfactuals estimated, how does the method interact with the model being explained, and what ground-truth knowledge is available for validation?
    b. The success rate varies drastically between 100% and 0% (Table 1 and the appendix), raising concerns about the reliability and consistency of the results.

---

> ### Author Response · Authors · 2025-02-02
>
> Dear reviewer AiGr,
>
> Thank you very much for reviewing our work.
>
> Re: The paper does not specify how the RL model is trained; How is policy model updated?
>
> >The concern that "the paper does not specify how the RL model is trained" or "how its learnable parameters are obtained" is central to Weaknesses 1, 2, and 3. However, this is clearly explained in Section 4 and Algorithm 1. Specifically, lines 24–26 in Algorithm 1 explicitly describe how the RL network parameters are updated. Additionally, Section 4 details how concepts in CFE correspond to the concepts in RL, and how searching for CFEs is formulated as an RL problem.
>
>
> Re: How are constraints incorporated in Line 13.
>
> >The incorporation of constraints is explained in the Constraints paragraph in Section 4.
>
>
> Re: Are candidate features selected one-by-one or in batches?
>
> >As stated in the Action paragraph in Section 4, each action is 3-dimensional: {time-step, feature, and strength}. Candidate features are selected one by one.
>
>
> Re: Is the approach computationally scalable?
>
> > By reducing the search space to 3-dimensional, our method scales effectively to multivariate time-series data. To our knowledge, no other CFE method without training datasets achieves this scalability for this data type.
>
>
> Re: The success rate varies drastically between 100% and 0%
>
> > As stated in the paper:
> > - "For RL-based baselines, CFRL and FastAR fail with a 0% success rate in 28/40 and 34/40 cases, respective"
> > - "CFRL and FastAR often fail to generate valid CFEs for complex multivariate time-series data. Please note that this is not a criticism of CFRL or FastAR, because they are not designed for multivariate time-series data."
> >
> > We included these baselines because they are also RL-based, providing a relevant comparison despite their limitations in handling multivariate time-series data.

---

> > ### Comment · Reviewer_AiGr · 2025-02-03
> >
> > **Dear Authors,**
> >
> > Thank you for your clarification.
> >
> > 1. **Learnable parameters in RL:** The statement **"No training data"** may be an overstatement or misleading, as the learnable parameters in **Lines 24–26 of Algorithm 1** still depend on data rather than being entirely data-free. It would be better to rephrase or provide a clearer explanation.
> >
> > 2. **Constraints:** Constraints are mentioned in **Line 14 of Algorithm 1** and **Section 4**, but their descriptions remain high-level and do not fully explain their incorporation. For example, **Section 4** states that **"SCM may be encoded into rules"**, but the specific types of rules and their integration into the feature selection process are not clearly defined. Providing additional details, such as an illustration or an example, would improve readability and enhance understanding.
> >
> > 3. **Candidate feature selection & computational cost:** Further clarification is needed regarding whether the model outputs:
> >    - A batch of **{time-step, feature, strength}** triplets at each step, or
> >    - **{time-step, a batch of features, strengths}**,
> >
> >     as selecting features one by one can still be computationally expensive, even when included in a triplet. This issue becomes particularly significant for large feature spaces—for example, an image with **1024 × 1024 × 3** pixels.
> >
> > Best regards

---

> > > ### Author Response · Authors · 2025-02-03
> > >
> > > Dear Reviewer AiGr,
> > >
> > > Thank you for the follow-up.
> > >
> > > Re: 1. The RL algorithm is not training-data free.
> > >
> > > > The data used in Algorithm 1 is testing data, i.e., the data to be explained or for which CFEs are generated. To clarify, baseline methods require a training dataset $D_{train}$ to generate CFEs for $X_{test}$, whereas our proposed method operates directly on $X_{test}$​ without the need for $D_{train}$.
> > > >
> > > > Thank you for the question. We will make it more clear.
> > >
> > >
> > > Re: 2. Constraints
> > >
> > > > The constraint can be applied straightforwardly. For instance, a constraint can be a rule that enforces that Feature 1 must remain within range [-1, 1]. If an action attempts to set Feature 1 to 2, the RL agent can either discard the change or adjust it to 1 instead.
> > >
> > >
> > > Re: 3. Candidate feature selection & computational cost
> > >
> > > > This work focuses on generating CFEs for multivariate time-series data without a training dataset. We acknowledge that the proposed method may not scale efficiently to scenarios with thousands of features. Extending our approach to such cases can be a direction for future work.
> > >
> > > Thank you!

---

### Review · Reviewer_9A2J · 2025-02-13

**Summary Of Contributions:**

This paper proposes No-Training-Dataset Counterfactual Explanation (NTD-CFE), a reinforcement learning (RL)-based method for generating counterfactual explanations (CFE) in both static and multivariate time-series data with continuous and discrete attributes. Notably, NTD-CFE operates without requiring a training dataset or similar data samples and is model-agnostic, making it compatible with any predictive model, including non-differentiable ones. Additionally, it allows users to specify which features can be modified and define dependencies between features, enabling the incorporation of feasibility constraints and causal relationships into the counterfactual generation process.

**Audience:**

Yes

**Broader Impact Concerns:**

There is no broader impact concern.

**Claims And Evidence:**

Yes

**Requested Changes:**

- Provide the motivations and further analysis for technical terms and specification.

- Supplement the related works as mentioned. If possible, empirically compare to the aforementioned approaches.

- Make the notions in reward and proximity more concise and easier to read. Rewrite the Reward and Proximity section.

**Strengths And Weaknesses:**

Strengths
- The idea of leveraging RL and counterfactual explanation is interesting.
- The proposed method is intuitive and can be applied to many data types.

Weaknesses
- The current writing needs to improve. Specifically, the paper only aims to present the technical specification the method, but lacks of providing their motivations and further analysis.
- The related works lack the discussions and references to other important works in explainable AI solving the same problem as this paper, typically [1], [2], [3]. It is encouraging to compare the proposed method and these methods.
- Some notions used in this paper are not good, leading to confusion, for example, the ones in Reward and Proximity. The notions need to be revised and this section needs to be rewritten to make it clearer.


[1] Ramaravind K Mothilal, Amit Sharma, and Chenhao Tan. 2020. Explaining machine learning classifiers through diverse counterfactual explanations. In Proceedings of the 2020 conference on fairness, accountability, and transparency.

[2] Annabelle Redelmeier, Martin Jullum, Kjersti Aas, and Anders Løland. 2021. MCCE: Monte Carlo sampling of realistic counterfactual explanations. arXiv preprint arXiv:2111.09790 (2021)

[3] Vo, V., Le, T., Nguyen, V., Zhao, H., Bonilla, E. V., Haffari, G., & Phung, D. (2023, August). Feature-based learning for diverse and privacy-preserving counterfactual explanations. In Proceedings of the 29th ACM SIGKDD Conference on Knowledge Discovery and Data Mining (pp. 2211-2222).

---

> ### Author Response · Authors · 2025-02-13
>
> Dear Reviewer 9A2J,
>
> Thank you for reviewing our work.
>
> Re: lack of motivation and further analysis
>
> > We provided motivation of the work in the Introduction section, especially "CFE methods for multivariate time-series data are less common" and "all existing model-agnostic CFE methods for multivariate time-series require access to large collection of samples from the training distribution of the model being explained. This requirement can be infeasible in real-world domains especially due to privacy and other concerns." This motivates the need of a CFE method for multivariate time-series data without a training dataset.
> >
> >
> > We provided analysis in section 4.1 and section 5, could you please be more specific about further analysis? We will revise the paper accordingly. Thank you.
>
>
> Re: Cite and compare with [1]  [2]  [3]
>
> > Thank you for the references. However, the 3 works are neither about multivariate time-series data nor operating without training datasets, which are the focus of our work.

---

> > ### Comment · Reviewer_9A2J · 2025-03-24
> > **My response**
> >
> > Thanks for your response and refining the paper. It is more readable now. Regarding [1,2,3], I suggested them because if with $K=1$, your approach can be applied to tabular data.

---

### Decision · Action_Editor_cExD · 2025-04-21

**Recommendation:** Accept with minor revision

**Comment:**

Using RL for counterfactual explanation is a straightforward and reasonable direction. The proposed method has been specifically designed the method in the scenario of time-series data.

The claim is supported by its clear design and analysis. All major issues have been resolved in the conversation between authors and reviewers.

The paper's contents are relevant to the TMLR audience's interest.

**Audience:**

The explanation is an important aspect of deep learning-based methods. The TMLR's audience with a general machine learning background might be interested in knowing a new method to tackle the explanation problem in a data-free manner.

The audience working on RL and Counterfactual explanations will be interested in the proposed method across these two domains.

**Claims And Evidence:**

The paper proposes a data-free counterfactual explanation method on time-series data. A reinforcement learning method has been specifically designed to solve this problem.

The claim, "RL-based data-free counterfactual explanation," is supported by the discussion and experimental analysis. The design of the RL algorithm is clearly stated in Section 4. The overall idea is straightforward and convincing despite some small issues about unclear descriptions and justifications. These issues could be solved by refining the paper's content organization.

The comprehensive experimental analysis demonstrated the effectiveness of the proposed method.

---

> ### Author Response · Authors · 2025-05-15
> **Thank You**
>
> Dear Action Editor and Reviewers,
>
> Thank you very much for the good news and for providing constructive feedback about our work!
>
> Best regards,
> Authors